# Regulation of nerve growth and patterning by cell surface protein disulphide isomerase

Geoffrey MW Cook[1], Catia Sousa[1,2], Julia Schaeffer[1], Katherine Wiles[1,3], Prem Jareonsettasin[1,4], Asanish Kalyanasundaram[1,5], Eleanor Walder[1,5], Catharina Casper[1,6], Serena Patel[1,5], Pei Wei Chua[1,7], Gioia Riboni-Verri[1,8], Mansoor Raza[9], Nol Swaddiwudhipong[1], Andrew Hui[1], Ameer Abdullah[1], Saj Wajed[1,10], Roger J Keynes[1]*

[1]Department of Physiology, Development and Neuroscience, University of Cambridge, Cambridge, United Kingdom; [2]Grenoble Institute des Neurosciences, La Tronche, France; [3]Independent researcher, London, United Kingdom; [4]Exeter College, Oxford, United Kingdom; [5]School of Clinical Medicine, Cambridge University Hospitals, Cambridge, United Kingdom; [6]Winter, Brandl, Fürniss, Hübner, Röss, Kaiser & Polte, Partnerschaft mbB, Patent und Rechtsanwaltskanzlei, München, Germany; [7]School of Medicine and Health Sciences, Monash University, Bandar Sunway, Malaysia; [8]School of Medicine, Medical Science and Nutrition, University of Aberdeen, Aberdeen, United Kingdom; [9]Cambridge Innovation Capital, Cambridge, United Kingdom; [10]University of Exeter Medical School, Exeter, United Kingdom

*For correspondence:
rjk10@cam.ac.uk

Competing interests: The authors declare that no competing interests exist.

**Abstract** Contact repulsion of growing axons is an essential mechanism for spinal nerve patterning. In birds and mammals the embryonic somites generate a linear series of impenetrable barriers, forcing axon growth cones to traverse one half of each somite as they extend towards their body targets. This study shows that protein disulphide isomerase provides a key component of these barriers, mediating contact repulsion at the cell surface in chick half-somites. Repulsion is reduced both in vivo and in vitro by a range of methods that inhibit enzyme activity. The activity is critical in initiating a nitric oxide/S-nitrosylation-dependent signal transduction pathway that regulates the growth cone cytoskeleton. Rat forebrain grey matter extracts contain a similar activity, and the enzyme is expressed at the surface of cultured human astrocytic cells and rat cortical astrocytes. We suggest this system is co-opted in the brain to counteract and regulate aberrant nerve terminal growth.

## Introduction

Peripheral spinal nerves have a striking anatomical periodicity, or segmentation, that reflects their necessary isolation from the segments of developing bone that will form the vertebral column. This study sets out to identify the molecular basis of this patterning. We find a critical role for the enzyme protein disulphide isomerase in separating outgrowing axons from the somite cells that generate the vertebrae, and provide evidence regarding the underlying mechanism (*Cook et al., 2019*).

In avian and mammalian embryos, both outgrowing motor and sensory axons, and migrating neural crest cells, encounter the periodic somites that flank both sides of the neural tube (future spinal cord). Here they traverse preferentially the anterior (A, rostral/cranial) - rather than posterior (P, caudal) - halves of each successive somite (*Keynes and Stern, 1984*; *Rickmann et al., 1985*; *Bronner-*

*Fraser, 1986*; *Fleming et al., 2015*; *Figure 1a*). For neural crest cells this preference has been shown to depend on repulsive signalling in the P-half-somite by members of the Semaphorin/Neuropilin- and Ephrin/Eph protein families (*Krull et al., 1997*; *Wang and Anderson, 1997*; *Koblar et al., 2000*; *Gammill et al., 2006*; *Davy and Soriano, 2007*; *Schwarz et al., 2009*). However the basis of axonal segmental patterning has remained elusive.

We previously identified contact repulsion as the main cellular mechanism generating axonal patterning (*Davies et al., 1990*; *Keynes et al., 1997*). Sequential repulsion of outgrowing motor and sensory axons in successive P-half-sclerotomes (future vertebrae) forces axons to traverse the anterior (A/cranial) halves. We showed that extracts of chick embryo somites cause growth cone collapse of both motor and sensory axons in vitro (*Davies et al., 1990*), a phenomenon that is widely used as a method for identifying molecules that regulate growth cone motility (*Kapfhammer and Raper, 1987*; *Raper and Kapfhammer, 1990*). Additionally we found that the lectins peanut agglutinin (PNA) and jacalin bind selectively to the surface of P-half-sclerotome cells rather than A-half-sclerotome cells (*Davies et al., 1990*; *Stern et al., 1986*). Immobilized PNA can be used to deplete collapse activity, and activity is recovered by lactose elution. Biochemical purification led to the

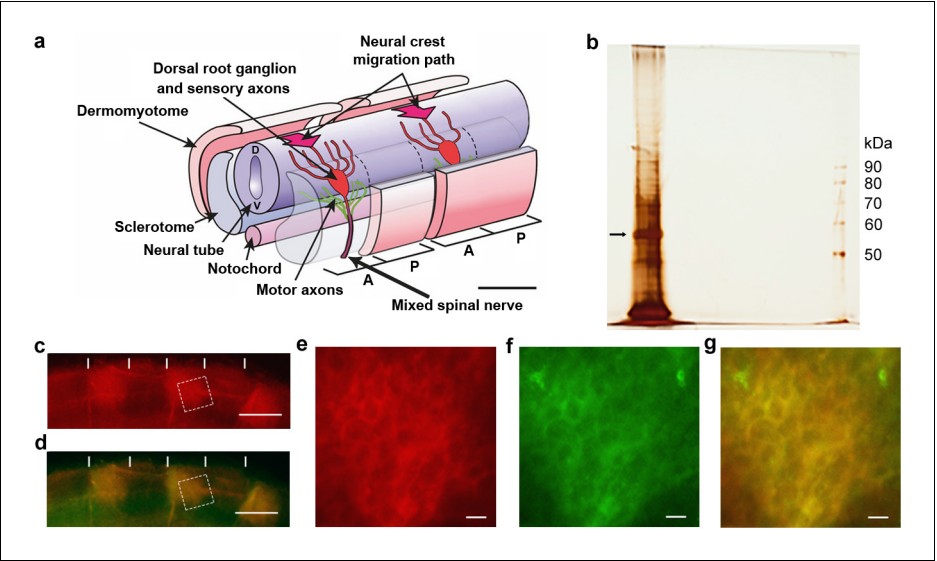

**Figure 1.** Identification of csPDI in somites. (a) Schematic diagram of two spinal nerve segments in the early embryo of birds and mammals. The neural tube (future spinal cord) extends longitudinally, overlying the midline notochord (future intervertebral discs); anterior/A (cranial/rostral) to the left and posterior/P (caudal) to the right. Two somites are shown, each sub-divided into 2 main components - the dermomyotome (future dermis and skeletal muscle) and sclerotome (future vertebral cartilage and bone). The sclerotome of each somite is further subdivided longitudinally into A- and P-halves (dashed lines), and the early components of the peripheral nervous system develop within the mesenchyme of each successive A-half-sclerotome. Here, motor axons (green) extend from cell bodies in the ventral (V) neural tube, and sensory axons (red) extend from cell bodies that coalesce to form the dorsal root ganglion (DRG). The DRGs derive from neural crest cells that earlier migrated into the A-half-sclerotomes (thick arrows) from the dorsal neural tube (D). Each DRG cell body generates one axon that grows dorsally to enter the neural tube and one that grows ventrally to join the motor axons; together with preganglionic autonomic axons (not shown), these motor and sensory axons form the 'mixed' spinal nerves that will innervate the body at each segmental level. Scale bar 50 μM. Adapted from *Kuan et al., 2004*. (b) Silver-stained SDS-PAGE of lactose eluate of chick somite proteins bound to PNA-agarose; arrow indicates the major band of 57 kDa. (c, d) Somite strip live-stained with rhodamine-PNA (red, (c) and co-stained with fluorescein-conjugated anti-PDI; preferential staining of three P-half-sclerotomes is shown; PNA staining and anti-PDI staining are co-localized (yellow, (d); vertical white lines indicate half-somite boundaries; Scale bars 50 μM. (e-g), Higher magnification of boxed regions in c and d showing ring staining at the cell periphery by rhodamine-PNA (e) and by fluorescein-conjugated anti-PDI (f), and their co-localisation (yellow, (g). Scale bars 5 μM.

The online version of this article includes the following figure supplement(s) for figure 1:

**Figure supplement 1.** Further characterisation of csPDI in somites.

identification of two PNA-binding glycoproteins shown by SDS-PAGE as two silver staining bands of 48 kDa and 55 kDa (*Davies et al., 1990*).

## Results

### Identification of cell surface PDI in somites

In the present work we combined PNA affinity purification with more effective inhibition of protease activity in the somite extracts, and examined the lactose eluates by semi-preparative SDS PAGE. This revealed a major silver-staining band of apparent molecular weight 57 kDa, closely matching the 55 kDa band seen in the earlier study (*Davies et al., 1990*; *Figure 1b*). The band was excised and submitted for tryptic digestion and mass spectrometry, revealing 57 peptides distributed throughout the extent of the enzyme protein disulphide isomerase/PDIA1/P4HB (*Figure 1—figure supplement 1a*).

PDIA1/P4HB is one of a PDI-family of proteins that share in common a thioredoxin-like structural fold (*Kozlov et al., 2010*). It is known principally as an intracellular enzyme localized in the endoplasmic reticulum (ER), where it regulates protein folding by catalyzing the formation and breakage of disulphide bonds (*Goldberger et al., 1963*; *Ali Khan and Mutus, 2014*; *Parakh and Atkin, 2015*). A PDIA1-related molecule was also identified previously as a retina-specific candidate cell adhesion molecule (*Hausman and Moscona, 1975*; *Pariser et al., 2000*). Our finding that somite cell surface PDI (csPDI) binds PNA, and is lactose-elutable from immobilized PNA, indicates that this form of PDI is O-glycosylated. This is supported by the observation that csPDI expressed by Jurkat T cells, immortalized from human T cell leukaemia, also possesses PNA-binding O-glycans, the elongation of which can be blocked experimentally (*Bi et al., 2011*; *Schaefer et al., 2017*). In addition, using a sensitive fluorescent reductase assay (*Raturi and Mutus, 2007*) we found that commercially purified (bovine liver) PDI does not bind to PNA-agarose, indicating that somite csPDI has an affinity for PNA based on its glycosylation state (*Figure 1—figure supplement 1b*). The expression of PDI at the surface of P-half-sclerotome cells was confirmed by live-staining of microdissected strips of chick somites, using both polyclonal anti-PDI antibody and fluorescently labelled PNA, which showed co-localization at the cell periphery in the P-half-sclerotome (*Figure 1c–g*). Also the onset of PNA staining in the P-half-sclerotome was found to precede the first emergence of motor and sensory axon outgrowth in the A-half-sclerotome by ~1.5–3 hr (*Figure 1—figure supplement 1c,d*).

### csPDI mediates spinal nerve patterning in vivo

A role for csPDI in mediating repulsion of outgrowing spinal axons in vivo was tested by siRNA knockdown of csPDI expression in chick embryo somites in ovo, predicted to promote outgrowth of motor and sensory axons into the P-half-sclerotomes. A construct was designed on the basis of the study of *Zai et al., 1999*. They used an antisense oligodeoxynucleotide directed against a 24 base pair target sequence in the 3' UTR of PDIA1/P4HB to show that csPDI expression in human erythroleukaemia cells is markedly reduced (>70%) without significantly affecting cell viability. The efficacy and specificity of this construct has also been shown by others (*Sobierajska et al., 2014*; *Janiszewski et al., 2005*). We initially confirmed that the chick siRNA construct inhibits expression of csPDI in primary cultures of chick retinal cells and P-half-sclerotome cells (*Figure 2—figure supplement 1a–d*). PDI gene knockdown in ovo was then carried out by microinjection of the siRNA, incorporated in a polyethylene glycol matrix, into at least 8 somites on one side of the embryo, anterior to the most recently formed somite (stage 9–12 *Hamburger and Hamilton, 1951*; *Figure 2a*). Confirmation of cs-PDI knockdown using Western blotting was not attempted due to the limiting availability of sufficient quantities of somite tissue, combined with the high ratio of constitutive PDI expression in the ER versus csPDI. As predicted however, PDI knockdown in ovo caused loss of extracellular PNA-binding in P-half-sclerotomes (*Figure 2—figure supplement 1e–g*). After siRNA injection and further incubation for 48 hr, spinal nerve outgrowth was assessed by immunohistochemistry using a neuron-specific-β-III tubulin antibody (clone TUJ1), observer-blind to treatment condition. Embryos treated with control/scrambled siRNA showed normal axon segmentation, with growth restricted to the A-half-sclerotomes (*Figure 2b*). However PDI knockdown caused outgrowing motor axons to project additionally into the P-half-sclerotomes adjacent to the neural tube/spinal cord (*Figure 2c,d*), an abnormal trajectory not seen in untreated embryos or in those similarly

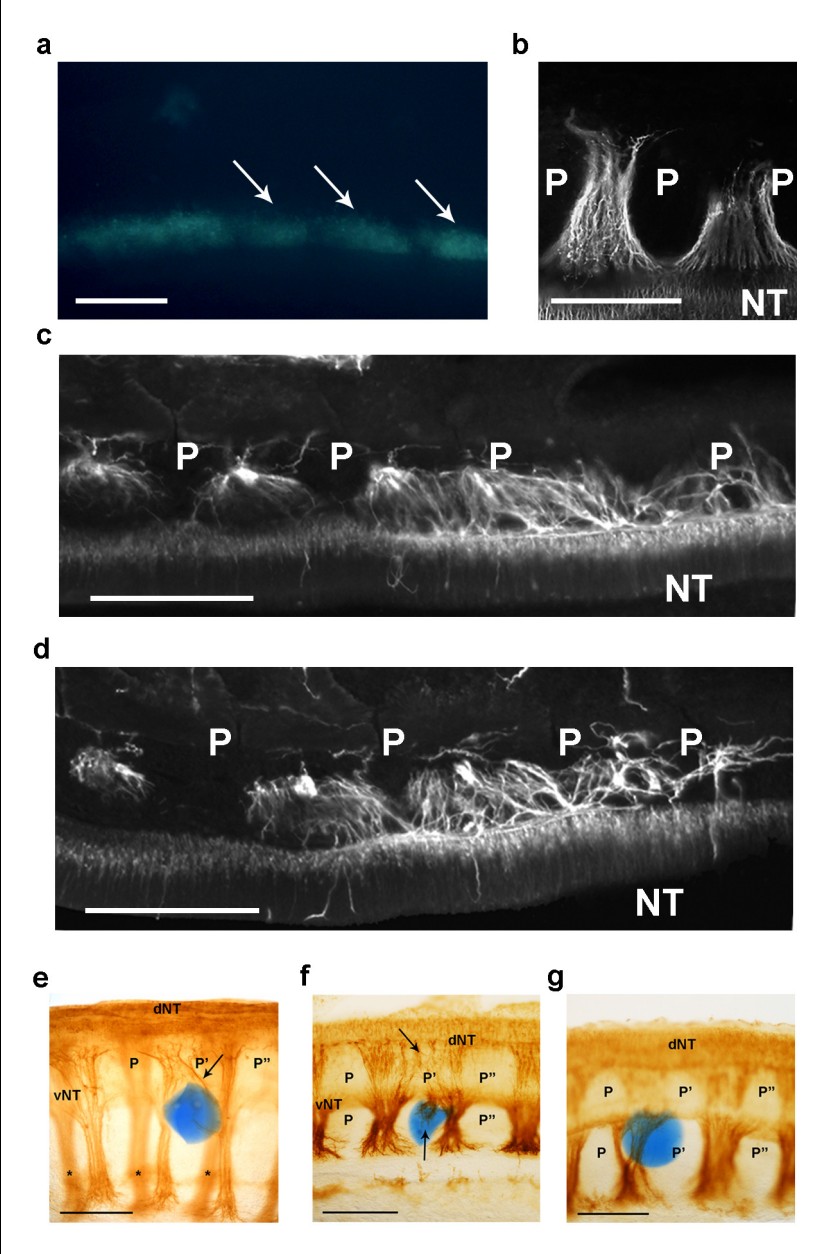

**Figure 2.** csPDI mediates nerve patterning in vivo. (**a**) Image of a live embryo in ovo, viewed from above, taken 24 hr after injection of fluorescein-labelled siRNA into somites (arrows) on one side; label is distributed throughout the A- and P-half-sclerotomes of each somite, and is visibly diminished, as expected, at 3 consecutive somite boundaries. Scale bar 100 µM. (**b**) Representative image of normal motor axon segmentation following scrambled siRNA delivery. Longitudinal section stained using fluorescein-conjugated TUJ1 antibody. Scale bar 100 µM. (**c, d**) Loss of axon segmentation in two embryos after PDI siRNA knockdown. The siRNA-treated side of each embryo is shown; axons are segmented normally (left) but this is disrupted (right) where axons grow into P-half-sclerotomes (P). NT, neural tube. Scale bars 100 µM. (**e, f**) Loss of axon segmentation in embryos after in ovo implantation of PACMA 31-impregnated bead (blue); embryos were stained using HRP-labelled TUJ1 antibody and viewed as whole-mounts (**e**) or as implanted-side-only half-mounts (**f**); abnormal growth of sensory axons (arrow, e; upper arrow, f) towards dorsal neural tube (dNT) in P-half-sclerotome (P′), compared with normal projections avoiding two adjacent P-half-sclerotomes (P, P′′); lower arrow (f) indicates motor axons sprouting from ventral neural tube (vNT) into P-half-sclerotome; asterisks, spinal axons on opposite side of whole-mount (**e**). Scale bars 150 µM. (**g**) Normal segmentation of dorsal/sensory axons and ventral/motor axons after implantation of PACMA 56 bead; P, P′, P′′, dorsal and ventral domains of 3 consecutive P-half-sclerotomes. Scale bar 150 µM.

The online version of this article includes the following figure supplement(s) for figure 2:

*Figure 2 continued on next page*

*Figure 2 continued*

**Figure supplement 1.** Phenotypic rescue of siRNA knockdown and effect of inhibiting csPDI using PACMA 31.

treated with control/scrambled siRNA. Control experiments showed that expression of the A-half-somite polarity determinant gene *Tbx18* was unaffected by siRNA injection, whereas expression of the P-half determinant gene *Uncx4.1* was variably diminished in the treated region (*Figure 2—figure supplement 1h*). Since *Tbx18* expression did not alter correspondingly, this was unlikely to be due to a P-to-A switch in cell identity, or to reflect a change in cell viability due to reduced PDI expression. It may be explained if csPDI knockdown in P-half-sclerotome cells at the A/P boundaries causes some to mix with neighbouring A-half cells and down-regulate *Uncx4.1* expression as a result. Injection of scrambled siRNA did not cause detectable sclerotome caspase-3 expression.

To rule out the possibility that these phenotypes resulted from an off-target effect of the siRNA, control experiments confirmed that co-injection of siRNA with a FLAG-M1-epitope-tagged plasmid expressing human PDIA1/P4HB (>90% homologous to chicken PDIA1 *Everson and Kao, 1997*) partially rescued the normal segmented axon phenotype (*Figure 2—figure supplement 1i–k*). We also found that inhibiting the enzyme activity of PDI caused a similar phenotype. PDI possesses two independent active sites, and the small molecule PACMA 31 has been shown to form a covalent bond with a cysteine residue of the second active site, thereby inhibiting its catalytic activity (*Xu et al., 2012*). PACMA 31 was applied in ovo using two delivery methods. First, as described above for siRNA delivery, PACMA 31 in solution (200 µM) was injected directly into somites in ovo and the resulting axon phenotype assessed by immunohistochemistry. PACMA 56, an inactive substituted alkynyl derivative of PACMA 31 that does not bind to PDI (*Xu et al., 2012*), acted as a control. Consistent with the results of siRNA knockdown, PACMA 31 injection also caused abnormal axon projections into P-half-sclerotome whereas control/PACMA 56 injection did not (*Figure 2—figure supplement 1l–n*). In addition the A-P width of ventral roots increased after PACMA 31 injection, indicating axon defasciculation (*Figure 2—figure supplement 1o*).

The second PACMA delivery method involved impregnation of Affi-Gel Blue agarose beads (25–50 µm diameter) with PACMA 31 or PACMA 56 (500µM), followed by microsurgical implantation of single beads in ovo between the neural tube and newly-formed sclerotome in stage 12–14 chick embryos. After further incubation for 24–36 hr, axon trajectories were assessed in the implant region by whole-mount immunohistochemistry. As with siRNA knockdown, PACMA 31 caused abnormal axon outgrowth into P-half-somite territory (*Figure 2e,f*; 14/29 embryos). Using PACMA 56 as control, only an occasional axon outgrowth abnormality (1/20 embryos) was observed; in 19/20 embryos, axons were confined to the A-half-sclerotomes as in normal embryos (*Figure 2g*).

## csPDI mediates axon repulsion via nitric oxide signalling/S-nitrosylation

To elucidate the mechanism of action of csPDI we first tested whether PDI causes growth cone collapse by direct interaction with the growth cone surface. The purified bovine enzyme incorporated in liposomes was added at a range of concentrations (25–1000 ng/ml) to cultures of chick embryo dorsal root ganglia (DRGs) extending sensory axons on laminin in the presence of nerve growth factor (NGF; typically between 50 to >100 growth cones were assayed per DRG). However this did not increase collapse above the control levels (0–20% of growth cones) seen after addition of phosphate-buffered saline (PBS) or untreated liposomes (*Figure 3a*).

Nitric oxide (NO) has been shown to elicit growth cone collapse in vitro when released in solution from NO donors [3-morpholino-sydononimine, SIN-1 (*Hess et al., 1993*); also 3-(2-hydroxy-1-methyl-2-nitrosohydrazino)-*N*-methyl-1-propanamine, NOC-7 (*Ernst et al., 2000*; *He et al., 2002*). Moreover *Zai et al., 1999* have shown that NO entry into csPDI-expressing human erythroleukemia cells involves a transnitrosation mechanism catalyzed by the enzyme. Physiological NO donor S-nitrosothiol (SNO) levels have been estimated in human cerebrospinal fluid and plasma at low micromolar concentrations (respectively 0.86 ± 0.04 µM *Bayir et al., 2003* and 1.77 ± 0.32 µM *Massy et al., 2003*). We therefore tested whether application of PDI in combination with S-nitrosoglutathione (GSNO, 1 µM) as NO donor causes growth cone collapse. Whereas application of GSNO alone in solution did not elicit collapse above control levels, significant collapse was observed when GSNO was first combined with PDI (125 ng/ml) and then added to DRG cultures (~60% growth cones

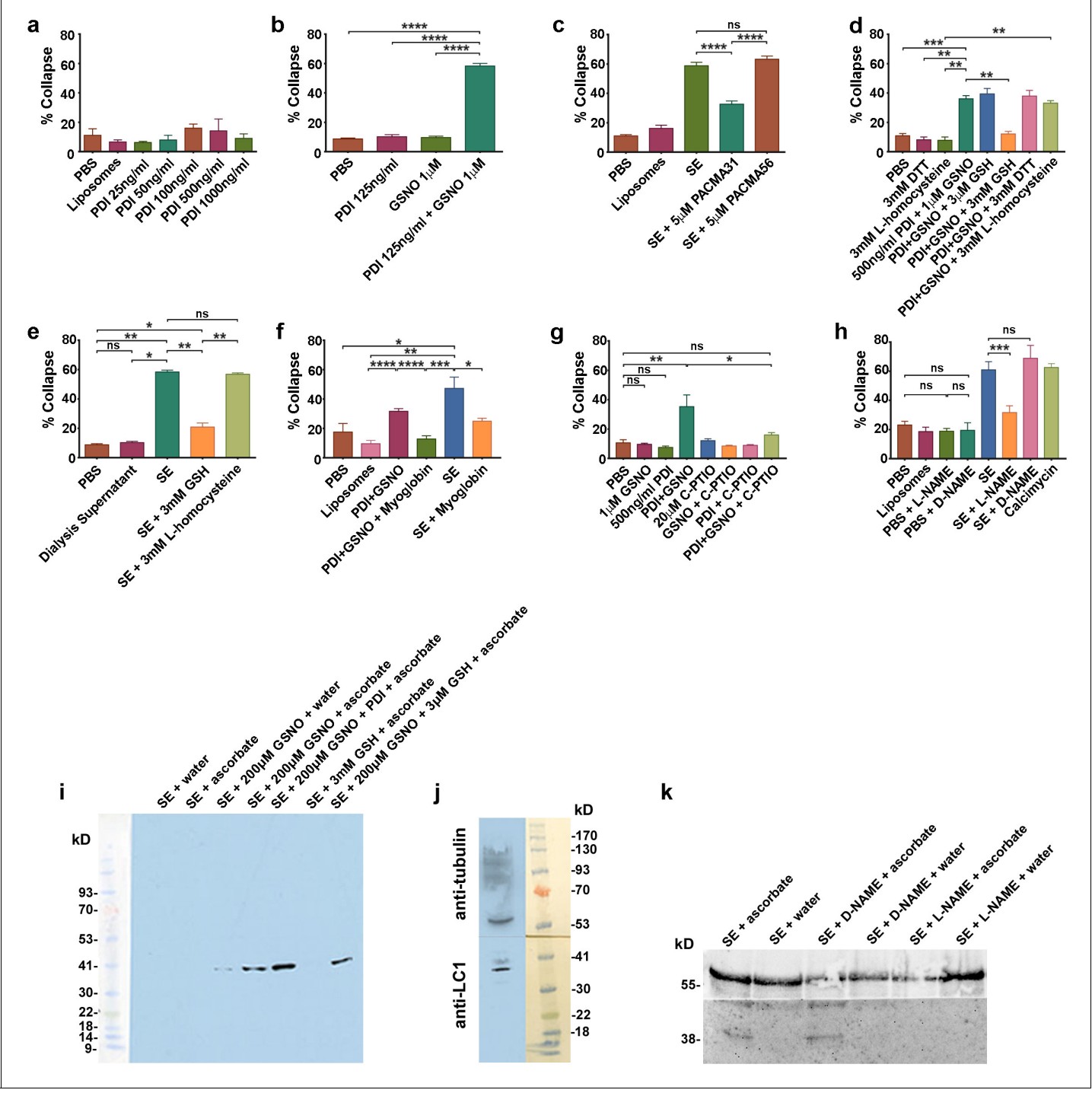

**Figure 3.** csPDI mediates axon repulsion in vitro. (**a**) Collapse assays testing purified bovine PDI in liposomes at a range of concentrations; controls, phosphate-buffered saline (PBS) and untreated liposomes; histogram shows mean + s.e.m. (**b**) Assays testing PDI and GSNO applied individually or concomitantly. (**c**) Assays testing PACMA31 and PACMA56 on somite extracts (SE). (**d**) Assays testing reducing agents at the concentrations indicated when applied either alone or together with PDI+GSNO. (**e**) Assays testing GSH and L-homocysteine on SE-induced collapse. (**f**) Assays testing myoglobin (20 µM) on SE- and PDI+GSNO-induced collapse. (**g**) Assays testing carboxy (C)-PTIO (20 µM) on PDI+GSNO-induced collapse. (**h**) Assays testing L-NAME and its control D-NAME on SE-induced collapse; calcimycin was used as a positive control. (**i**) S-nitrosylated protein (iodo-TMT-labelled) in somites; protein samples (48 µg) from somite cell-free extracts were fractionated on NuPAGE 4–12% Bis Tris gels as described in the Methods; lanes 1 and 2 are controls consisting of somite proteins only, with no detectable signal compared with lanes 3 and 4 where GSNO has been added; lane 3 is a control (treated with water) showing negligible iodoTMT labelling, and lane 4 (reduced with ascorbate to generate a new free thiol for labelling) shows increased label; lane 5 shows that addition of PDI (1 µg/0.1 ml reaction mixture) enhances labelling; lane 6 shows that 3 mM GSH in

*Figure 3 continued on next page*

*Figure 3 continued*

the absence of GSNO and PDI does not generate a signal; lane 7 shows that 3 µM GSH is insufficient to interfere with nitrosylation, concurring with the findings of *Sliskovic et al., 2005*. The coloured molecular weight markers on the blot are shown on the left (BLUeye prestained protein ladder, 2.5 µL, Geneflow). (j) Identification of LC1 in somite extract (25 µg protein); the blot was cut in half above the 41K marker and the top half of the membrane was probed with rabbit anti-tubulin followed by goat anti-rabbit IgG; the bottom half was probed with mouse monoclonal antibody against amino acids 2257–2357 of mouse MAP-1B (LC1) followed by goat anti-mouse IgG. The molecular weight markers on the blot are shown to the right (BLUeye prestained protein ladder, 3 µl). (k) Identification of LC1 as a substrate for S-nitrosylation; cell-free somite extract (200 µg) was treated with D-NAME or L-NAME, followed by further incubation in GSNO (200 µM), as described in the Methods. Samples were then processed for the presence of S-nitrosylated proteins using iodo-TMT as described in the Methods. Protein samples (15 µg) were then fractionated and blotted, after which the blot was cut as described for (j). The top half was treated with anti-tubulin and the bottom half with anti-iodoTMT. L-NAME treatment blocked S-nitrosylation, as shown by the lack of iodoTMT labelling. The control D-NAME was without effect.

The online version of this article includes the following figure supplement(s) for figure 3:

**Figure supplement 1.** Additional characterisation of csPDI-mediated growth cone collapse.

---

collapsed after 1 hr, *Figure 3b*, *Figure 3—figure supplement 1a,b*). To confirm that PDI+GSNO-induced collapse in solution is reproduced in the liposome-based collapse assay, we found that the PDI concentration dependence of collapse was similar in both cases. (*Figure 3—figure supplement 1a,c*). Also the time course of PDI+GSNO-induced collapse was similar to that induced by somite extracts, and contrasted with the more rapid onset of collapse induced by the soluble repellent Sema3A (*Figure 3—figure supplement 1d–f*).

We next tested the PDI inhibitors purified-bacitracin (*Rogelj et al., 2000*), anti-PDI neutralizing antibody, phenylarsine oxide (PAO) (*Bennett et al., 2000*), acetylated triiodothyronine (T3) (*Primm and Gilbert, 2001*) and 16F16 (*Hoffstrom et al., 2010*) on PDI+GSNO-induced collapse when applied in solution. Of these, three inhibitors (bacitracin, neutralizing antibody and PAO) were most effective in reducing collapse when incorporated in liposomes with PDI (*Figure 3—figure supplement 1g–l*). Following the publication of the small molecule PDI-inhibitor PACMA 31 and its control PACMA 56 (*Xu et al., 2012*), the candidacy of csPDI in mediating somite extract (SE)-induced collapse was further confirmed using PACMA 31 in liposomes, which inhibited collapse by >50% whereas PACMA 56 was inactive (*Figure 3c*).

PDI has two active sites, each with the amino acid sequence WCGHCK. *Sliskovic et al., 2005* have shown that PDI can be S-nitrosylated (PDI-SNO), and that the enzyme can also act as a denitrosylase resulting in •NO release as a free radical. They have proposed that PDI-SNO is denitrosylated by a one-electron reduction mechanism at the second active site. Moreover they showed that glutathione (GSH) is the most effective reducing agent, and that no significant denitrosylation is observed using reducing agents dithiothreitol (DTT) or L-homocysteine (*Sliskovic et al., 2005*). Consistent with their study, we found that when PDI+GSNO or somite extracts were incorporated in liposomes and subsequently treated with GSH (3 mM), collapse activity was lost. However identical experiments using 3 mM DTT or L-homocysteine did not affect collapse activity. Notably, GSH did not block collapse when applied at 3 µM, within the extracellular concentration range typically present in vivo and 3 orders of magnitude below the ambient intracellular concentration range (*Owen and Butterfield, 2010*; *Figure 3d*; see Discussion). Also consistent with the observations of *Sliskovic et al., 2005*., somite extract-induced collapse was inhibited by 3 mM GSH but not by 3 mM L-homocysteine (*Figure 3e*).

Further evidence that a NO-based mechanism elicits growth cone collapse was provided by the finding that myoglobin, regarded as a pseudo-enzymatic NO scavenger (*Ascenzi and Brunori, 2001*; *Rayner et al., 2009*), inhibited collapse induced by PDI+GSNO and by somite extracts (*Figure 3f*). PDI+GNSO-induced collapse was also inhibited by the membrane-impermeable NO-scavenger carboxy-2-Phenyl-4,4,5,5-tetramethylimidazoline-1-oxyl 3-oxide (C-PTIO; *Figure 3g*). Somite extract-induced collapse was additionally prevented by pre-treatment of DRGs with the neuronal nitric oxide synthase (nNOS) inhibitor L-NAME, but not by its control/chiral isomer D-NAME, indicating a role for nNOS activity in the collapse process (*Figure 3h*).

These experiments implicate NO signalling in somite-induced growth cone repulsion. The in vitro study of *Stroissnigg et al., 2007* has likewise shown a role for S-nitrosylation of the microtubule-associated protein MAP1B in mediating mouse DRG growth cone collapse caused by stimulation of nNOS by the calcium ionophore calcimycin/A23187. They showed further that S-nitrosylation of Cys

2457 in the MAP1B light chain sub-unit (LC1) is a critical event in the cytoskeletal dynamics underlying collapse. To examine expression of S-nitrosylated proteins in somites, we therefore carried out Western blotting of cell-free somite extracts using iodoTMT reagent, which gives lower background labelling compared with biotin labelling. The assay was prepared from 650 dissected somite strips homogenised in HENS buffer. Remarkably only one major band (molecular weight 38 kDa) was detectably S-nitrosylated (*Figure 3i*). We additionally confirmed, by Western blotting using a mouse monoclonal antibody against amino acids 2257–2357 in LC1 of mouse MAP-1B (see Materials and methods), that this S-nitrosylated somite protein reacts strongly with the anti-LC1 antibody (*Figure 3j*). Also the nNOS inhibitor L-NAME inhibited both S-nitrosylation of the 38 kDa protein (*Figure 3k*) and somite extract-induced growth cone collapse (*Figure 3h*), while the control stereoisomer D-NAME was without effect (*Figure 3h,k*). These findings indicate that a molecular mechanism similar to that proposed by *Stroissnigg et al., 2007* operates within the growth cone during its repulsion by somites in vivo.

## csPDI activity in mammalian forebrain grey matter

We previously found that, as for somites, extracts of adult mammalian and chicken forebrain grey matter also cause sensory/DRG axon growth cone collapse that can be depleted by the use of immobilized PNA. This suggested that a contact-repulsive system similar to that in somites may be expressed in the mature CNS (*Keynes et al., 1991*). In confirmation we found that immobilized jacalin, a lectin that binds the same O-linked disaccharide (Galβ1-3GalNAc) as does PNA, but unlike PNA is not selective for its de-sialylation, can be used to deplete collapse induced by rat forebrain extracts (RFE; *Figure 4—figure supplement 1a*). *Ghosh and David, 1997* have also described a growth cone collapse-inducing activity in membrane preparations of rat cerebral cortical grey matter. We therefore tested whether, as for somite extracts, a range of inhibitors of PDI activity block RFE-induced collapse, and found this was the case. Application of PACMA 31 (5μM) significantly reduced collapse (by 50–60%) whereas PACMA 56 (5μM) did not (*Figure 4a*, *Figure 4—figure supplement 1b*). Another small molecule PDI-inhibitor, quercetin-3-O-rutinoside (*Jasuja et al., 2012*) inhibited RFE-induced collapse when used at both 1 μM and 50 μM (*Figure 4b*). RFE-induced collapse activity was immunodepleted using polyclonal anti-PDI antibody but not using IgG or bovine serum albumin (BSA) as sham protein controls (*Figure 4c*). Moreover, as for somite extracts, application of 3 mM GSH reduced RFE-induced growth cone collapse, whereas 3 mM DTT, 3 mM L-homocysteine (L-HC) or 3 μM GSH did not (*Figure 4d*). At GSH concentrations between 3 μM and 3 mM, inhibition of collapse increased with concentration (*Figure 4—figure supplement 1c*). Last, and again consistent with a NO-based mechanism, we confirmed that application of the NO scavengers myoglobin and C-PTIO depleted RFE-induced collapse (*Figure 4—figure supplement 1d,e*).

One source of csPDI in the brain may be the astrocyte, which shares fate specification by the transcription factor SOX9 with P-half-sclerotome cells (future vertebral cartilage) (*Akiyama et al., 2002*; *Stolt et al., 2003*; *Sun et al., 2017*). In support of this, live-staining experiments showed that csPDI is expressed on the surface of cultured human astrocytic cells, as for P-half-sclerotome cells (*Figure 4e*). Moreover extracts of these cells caused growth cone collapse that was removed by the use of immobilized PNA and jacalin (*Figure 4—figure supplement 1f*), and a cell surface preparation isolated from rat primary cortical astrocytes was found to contain csPDI (*Figure 4—figure supplement 1g*).

Collectively these experiments indicate that csPDI is a major component of the growth cone collapse-inducing activity detectable in the grey matter of the mature mammalian brain. The NGF-dependent primary sensory neurons assessed here project axons in vivo in the CNS as well as PNS, making synapses in the dorsal horn of the spinal cord and in the brainstem. We have shown previously that two populations of CNS-restricted neurons are also responsive to the somite contact-repellent system. When explants of embryonic day-4 (E4) chick telencephalon or E7 retina are grafted in ovo in place of chick spinal cord, their axons avoid P-half-somites (*Keynes et al., 1991*; *Vermeren et al., 2000*). Moreover chick retinal axon growth cones collapse in response to somite extracts in vitro (*Vermeren et al., 2000*), and in further confirmation we found that they collapse in response to PDI+GSNO (*Figure 4—figure supplement 1h*).

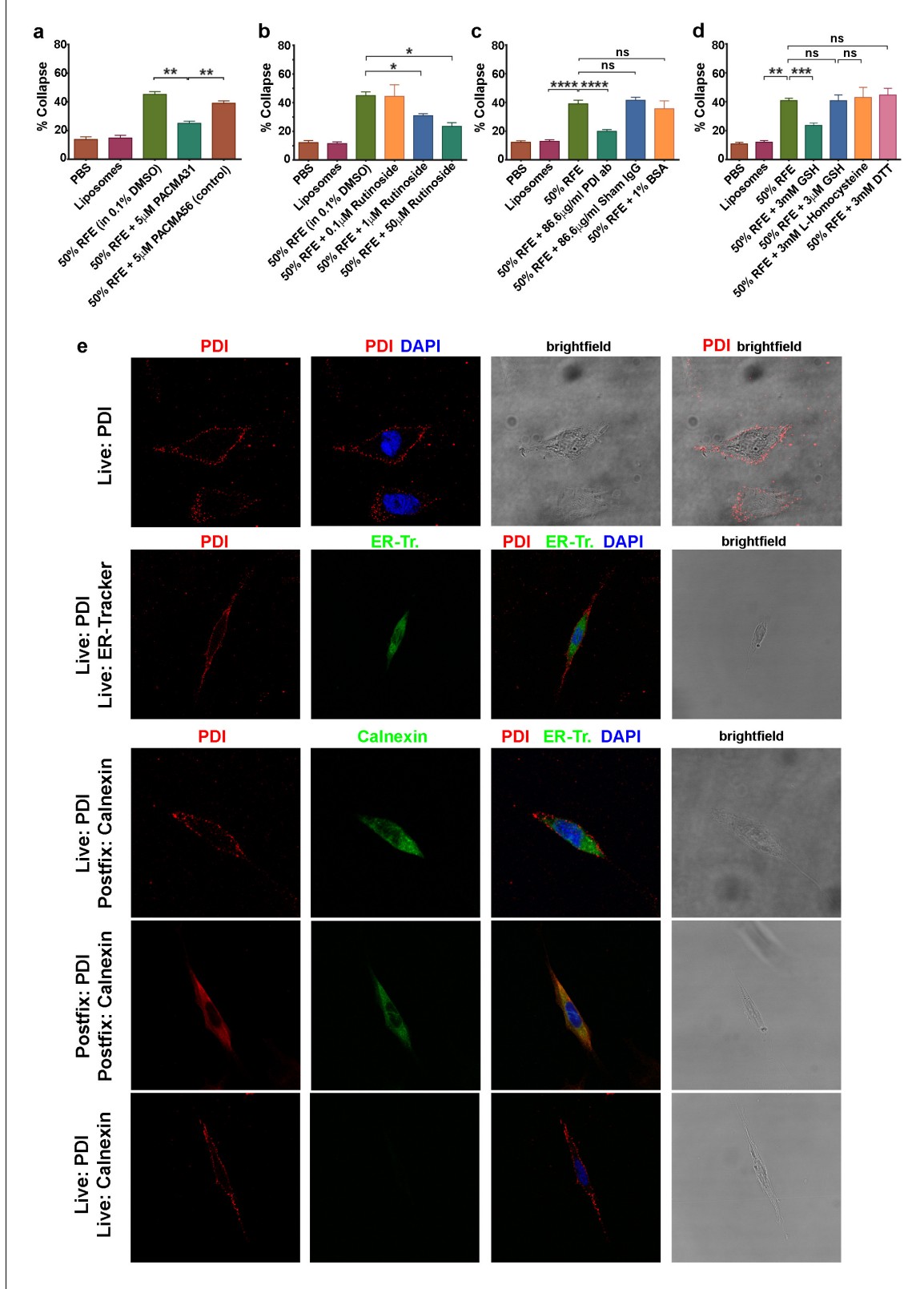

**Figure 4.** csPDI activity in mammalian forebrain. (**a-d**) Collapse assays using PACMA 31 and PACMA 56 (**a**), quercetin-3-O-rutinoside (**b**), immunodepletion (**c**), and reducing agents (**d**). (**e**) Immunocytochemistry on live human astrocytic (1718) cells. Scale bars 20 μM. Row 1, anti-PDI (red) shows PDI expression at the cell surface, DAPI-staining (blue) shows position of nucleus. Row 2, anti-PDI live staining at the cell surface (red) contrasts

*Figure 4 continued on next page*

*Figure 4 continued*

with selective staining of ER with ER-Tracker (green). Rows 3,4, fixation permits visualisation of ER-PDI using anti-calnexin (green), which is absent under live staining conditions (Row 5).

The online version of this article includes the following figure supplement(s) for figure 4:

**Figure supplement 1.** Action of lectins and other reagents on collapse activity in brain/astrocyte extracts and responsivity of retinal neurons.

## Discussion

The significance of PDI as an ER-based foldase/isomerase is well known, but the biological role of csPDI is less clear-cut. It has been implicated in processes such as platelet aggregation and thrombosis, and in animal cell infection by a variety of micro-organisms (*Ali Khan and Mutus, 2014*). Here we have identified a key function for csPDI in contact repulsion using a biological system. Consistent with its location at the cell surface, somite csPDI is an O-glycosylated protein, as shown by our lectin-binding studies (*Davies et al., 1990*) and by Bi et al. for human Jurkat T cells (*Bi et al., 2011*). In keeping with this, it has been shown that contact between a single DRG growth cone filopodium and the surface of a P-half-somite cell in vitro is sufficient to initiate a rapid filopodial withdrawal/repulsive response, followed by reorientation of the growth cone away from the cell (*Steketee and Tosney, 1999*). The rapid nature of this response, combined with our finding that somite extract collapse-inducing activity is depleted using the lectins PNA and jacalin (*Davies et al., 1990*), argue strongly that repulsion is likely to arise from the activity of csPDI rather than ER-based PDI. These lectins have specificity for O-glycans that are synthesised and linked to protein in the Golgi apparatus before the glycoprotein is transported to the cell surface. Also consistent with a repulsion mechanism, inhibition of PDI activity in vivo, using either siRNA knockdown or PACMA 31 inhibition of enzyme activity, causes axons to traverse the P-half-somites. Since PDI is a multifunctional enzyme operating both within and outside cells, it is possible that this phenotype might arise for other reasons. However we used the same target for gene knockdown experiments as used by *Zai et al., 1999.*, who showed that cell viability is unperturbed despite inhibition of csPDI expression. Additionally we saw no change in somite morphology despite loss of lectin binding at the cell surface.

In chick somites the onset of csPDI expression in P-half-sclerotome immediately precedes the first emergence of spinal axons in the A-half. This matches well the proposed function of csPDI in mediating contact repulsion of outgrowing motor and sensory axons. The selective migration of neural crest cells in A-half-sclerotomes precedes by several hours the first axon outgrowth at each segmental level in the chick embryo, and is likewise matched temporally by the onset of expression of the secreted repellent Sema3F in newly-formed P-half-sclerotomes (*Gammill et al., 2006*). The secreted axon repellent protein Sema3A is additionally expressed selectively in P-half-sclerotome (*Eickholt et al., 1999*) (see also *Shepherd et al., 1996*). However no long-distance axon repulsion is detected in collagen gel co-explants of DRGs with dissected P-half-sclerotomes (*Keynes et al., 1997*). Moreover, segmented spinal nerve patterning persists in compound Neuropilin1/2 mutant mice in which somite-based Semaphorin signalling is depleted (*Schwarz et al., 2009*; *Huber et al., 2005*), presumably because csPDI expression in these mice compensates disrupted neural crest migration.

Together these observations imply that the segmental patterning of neural crest cells and axons is regulated predominantly by distinct molecular signals. Supporting this, the Eph-family receptor tyrosine kinase EphB2 and its ligand ephrin-B1 have been additionally implicated in somite-based repulsion of neural crest cells (*Krull et al., 1997*; *Wang and Anderson, 1997*; *Davy and Soriano, 2007*), but shown not to be necessary for motor axon segmentation (*Koblar et al., 2000*). Other candidate axon-repellent molecules that are preferentially expressed in P-half-sclerotome have been identified (*Kuan et al., 2004*) but their in vivo roles have remained uncertain. For T-cadherin (*Ranscht and Bronner-Fraser, 1991*) and F-Spondin (*Tzarfati-Majar et al., 2001*), while each of these proteins has been shown to inhibit motor axon growth in vitro, mouse gene knockout phenotypes consistent with a role in spinal nerve segmentation in vivo have not been published to date (*Ciatto et al., 2010*; *Palmer et al., 2014*). The chondroitin sulphate proteoglycans aggrecan and versican provide further similar examples (*Watanabe and Yamada, 2002*; *Perissinotto et al., 2000*; *Dours-Zimmermann et al., 2009*). Also the extracellular matrix glycoprotein Fibulin 2 was recently found to have P-half-specific expression in a RNA-seq screen of dissected mouse A- and P-half-

somites (*Schaeffer et al., 2018*). While Fibulin 2 does not possess intrinsic growth cone collapse-inducing activity, the evidence indicates that it promotes Sema3A signalling in the P-half-sclerotome, and may contribute to spinal nerve fasciculation in vivo (*Schaeffer et al., 2018*). Last, the identification by Pfaff and colleagues of the *Columbus* mouse mutation (*Bai et al., 2011*) provides a striking example of loss of motor/sensory axon repulsion in mouse P-half-somites. Here, the loss of Presenilin-1 (PS-1) function causes outgrowing axons to traverse both halves of the somite. This is explicable by the consequent loss of *Notch* function required to generate P-half-somite polarity, upstream of P-half-somite differentiation, as seen in two previous PS-1 mouse gene knockout studies (*Shen et al., 1997*; *Wong et al., 1997*).

Regarding the mechanism of csPDI, the models of *Zai et al., 1999* (using human erythroleukaemia cells), *Ramachandran et al., 2001* (using fibroblasts and endothelial cells) and *Sliskovic et al., 2005* have been proposed to explain how NO entry into these cells is regulated by a transnitrosation mechanism facilitated by csPDI. These in vitro cellular models are directly applicable to the axon growth cone/somite system in vivo. We suggest that csPDI acts as a de-nitrosylase, operating constitutively at the P-half-somite cell surface to promote the transfer of NO• from extracellular S-nitrosothiols into the cytosol of contacting growth cone filopodia, thereby initiating repulsion/collapse

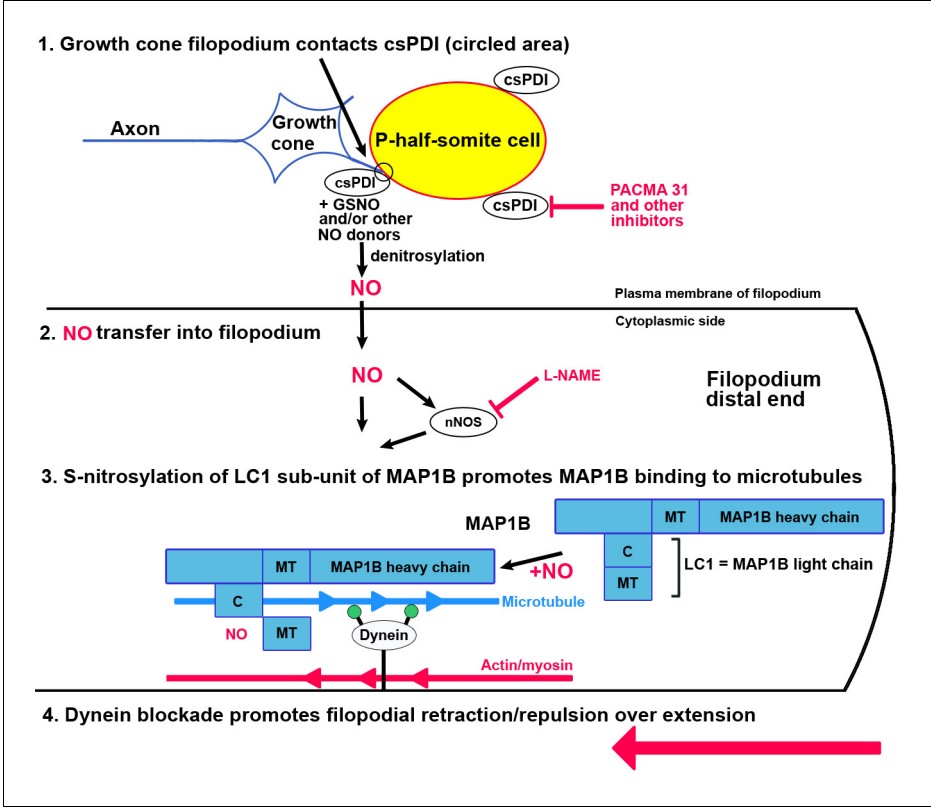

**Figure 5.** Schematic diagram of the proposed action of csPDI in mediating growth cone repulsion by NO. 1. A filopodium (blue) contacts a sclerotome cell in the posterior/P-half-somite, where it encounters csPDI, leading to the denitrosylation of a NO-donor molecule, for example GSNO. This reaction can be blocked by PACMA 31, so preventing collapse. 2. NO (which is known to cross cell membranes) is transferred into the filopodium as a result, where it may additionally regulate nNOS activity (*Benhar et al., 2009*). 3. As shown by *Stroissnigg et al., 2007*., nNOS activation causes nitrosylation of cysteine 2457 in the COOH terminus of the MAP1B light chain LC1. They propose that the resulting conformational change in LC1 enhances MAP1B/microtubule binding, which in turn blocks activity of the anchored motor protein dynein. Dynein normally provides a tubulin/minus-end-directed motor activity via its motor domains (filled green circles) that drives axon extension through interaction with microtubules. This is counteracted by a retraction force due to cortical actin/myosin, so that dynein blockade converts filopodial extension to filopodial retraction/repulsion (large red arrow). MT, microtubule-binding domain of MAP1B/LC1; C, COOH terminus of LC1. The lower part of this diagram is adapted from the scheme of *Stroissnigg et al., 2007*.

(*Figure 5*). Extracellular S-nitrosoglutathione (GSNO) may provide a ubiquitous source of NO donor, as suggested for csPDI activity in the context of platelet aggregation (*Shah et al., 2007*). Critically we have shown that its ambient extracellular concentration in vivo (~1 μM) is sufficient, in combination with purified PDI, to elicit growth cone collapse in vitro. Alternatively or additionally, other NO donors may be involved.

The transnitrosation model is well supported by the finding that DRG/sensory axon growth cones collapse when exposed in vitro to the NO donors 3-morpholino-syndoninime (SIN-1) (*Hess et al., 1993*) and NOC-7 (*Ernst et al., 2000*; *He et al., 2002*). SIN-1-induced collapse is prevented by the presence of haemoglobin, which binds released NO (*Hess et al., 1993*), and we have shown that both myoglobin, which similarly binds released NO, and the membrane-impermeable NO-scavenger PTIO (*Flögel et al., 2001*) deplete the collapse-inducing activity of somite extracts. Moreover nNOS inhibition by L-NAME prevents such collapse, indicating that NO signalling may be further amplified in the growth cone by nNOS activity (*Benhar et al., 2009*).

How might NO signalling in the growth cone influence the cytoskeleton? Our results accord well with the proposal of *Stroissnigg et al., 2007* that, in axons extending in vitro, growth cone retraction is counteracted by a microtubule/dynein-based system. S-nitrosylation of LC1 induces a conformational change that enhances binding of the LC1-HC complex to microtubules, so blocking dynein action and promoting retraction over extension. Correspondingly, in the somite system in vivo both motor and sensory axon growth cones extending at the A/P-half-somite boundaries will make filopodial contact with P-half-somite cells expressing csPDI, triggering NO-mediated repulsive signalling. Consistent with this, using the iodo-TMT reagent we find that LC1 is the only S-nitrosylated protein detected in cell free extracts of dissected somite strips, which necessarily include growth cone proteins.

The observation that a repellent activity closely similar to the somite system is expressed in mammalian forebrain grey matter is of particular interest, and extends the range of brain proteins originally identified as developmental axon repellents (*Raper and Kapfhammer, 1990*; *Cox et al., 1990*). Collapse-inducing activity is significantly reduced using the lectins PNA and jacalin (this study and *Keynes et al., 1991*), and is also prevented using a variety of small molecule inhibitors (PACMA 31, rutinoside, GSH) as well as myoglobin and anti-PDI antibody. Our findings additionally implicate the astrocyte as a source of this activity, since human astrocytic (1718) cells and rat cortical primary astrocytes express csPDI, and 1718-cell-derived growth cone collapse activity is removed by immobilized PNA and jacalin. In view of the involvement of NO/S-nitrosylation signalling in the csPDI-mediated repulsion mechanism, rather than a protein-based ligand-receptor interaction, a broad range of CNS axon types may prove susceptible to it. And consistent this, we have shown previously that chick CNS (retinal and telencephalic) axons respond to the somite repellent in vivo (*Keynes et al., 1991*; *Vermeren et al., 2000*). It may also be significant that csPDI expression by human malignant glioblastoma cells has been related to their invasiveness within the brain (*Goplen et al., 2006*).

The neuron may be another source of brain-derived csPDI, since a recent proteomic analysis of CNS synaptic cleft proteins identified csPDI/P4HB among the most enriched candidates at both excitatory and inhibitory synapses in embryonic rat cortical neuronal cultures *Loh et al., 2016*; csPDI has also been identified at the surface of both neuroblastoma cells (*Xiao et al., 1999*) and retinal cells (*Pariser et al., 2000*). Together with the experiments reported in this study, these findings collectively raise the possibility that csPDI is 'bifunctional' in promoting both adhesive and repulsive neuronal/glial interactions in the CNS. For example, NO signalling is implicated in synapse elimination during CNS development (*Wu et al., 1994*; *Gibbs and Truman, 1998*), and csPDI activity might provide an extracellular source of NO alongside intracellular nNOS activity.

In sum, this study reveals a novel role for the multifunctional enzyme PDI in the periodic patterning of peripheral spinal nerves, ensuring their separation in somites from developing cartilage and bone. The additional expression of csPDI at the astrocyte surface, and its function in promoting NO-based repulsion of growing nerve terminals, suggest a promising candidate for regulating axon growth and plasticity that may be widely distributed in the developing and mature nervous system.

# Materials and methods

**Key resources table**

| Reagent type (species) or resource | Designation | Source or reference | Identifiers | Additional information |
|---|---|---|---|---|
| Cell line (*Homo-sapiens*) | Human astrocytoma-derived 1718 cells | ATCC | RRID:CVCL_1118 | authentication by STR profiling mycoplasma contamination not tested (used directly from ATCC) |
| Transfected construct (*Gallus gallus*) | siRNA to 3'UTR of chick P4HB/PDIA1 | This paper | siRNA | TCGCCCTCAC TTGTCTTTA |
| Transfected construct (*Gallus gallus*) | scrambled siRNA | This paper | siRNA | GCTCTCTCG TCTATCTACT |
| Biological sample (*Gallus gallus*) | somite extract | This paper | | Freshly isolated from Gallus gallus |
| Biological sample (*Rattus norvegicus domestica*) | rat forebrain extract | This paper | | Freshly isolated from Rattus norvegicus domestica |
| Antibody | Mouse IgG2a anti-tubulin β3 (mouse monoclonal) | BioLegend | clone TUJ1 RRID:AB_2315519 | (1:500) |
| Antibody | anti-PDI (rabbit polyclonal) | Sigma-Aldrich | Cat# P7496 | (1:250 live staining) (1:500 post-fixation staining) (1:20,000 western blot) |
| Antibody | anti-calnexin mouse IgG2b (mouse monoclonal) | Abcam | clone 6F12BE10 RRID:AB_10860712 | (1:100 live staining) (1:200 post-fixation staining) |
| Antibody | Alexa Fluor 488 Goat anti-Mouse IgG1 (goat polyclonal) | Invitrogen | A-21121 RRID:AB_2535764 | 1:500 |
| Antibody | Alexa Fluor 594 Goat anti-Mouse IgG2a (goat polyclonal) | Invitrogen | A-21135 | 1:500 |
| Antibody | Peroxidase-conjugated Goat anti-mouse IgG (goat polyclonal) | Jackson Immuno Research | 115-035-003 | 1:500 |
| Recombinant DNA reagent | plasmid encoding human PDIA1 (18–508) | Addgene | hPDI1_18–508_WT_pFLAG-CMV1 RRID:Addgene_31382 | Depositing lab: Prof David Ron University of Cambridge, UK |
| Sequence-based reagent | *Uncx4.1* | This paper | primer synthesis | (forward) ATCGATGGA TTACTGAGCGG (reverse) TAATACGA CTCACTATA GGGAGGTTTAAGC AAACGGACGCTG |

*Continued on next page*

*Continued*

| Reagent type (species) or resource | Designation | Source or reference | Identifiers | Additional information |
|---|---|---|---|---|
| Sequence-based reagent | *Tbx18* | This paper | primer synthesis | (forward) GTAATGCT GACTCCCCGGTA (reverse) TAATACGACTCAC TATAGGGAGACTGG TTTGGTTTGTGAGCC |
| Sequence-based reagent | T7 promoter | Sigma-Aldrich | primer synthesis | TAATACGACTCA CTATAGGGAG |
| Commercial assay or kit | In-Fusion HD Cloning | Clontech | 639647 | |
| Other | ER-Tracker Green FL Glibenclamide | Thermo Fisher | E34251 | |

## Chick embryo grafting procedure

Fertilized hens' eggs (Gallus gallus, Bovans Brown variety; Winter Egg Farms, Fowlmere, Cambridgeshire) were incubated at 38℃ to obtain embryos at stage 12–14 (*Hamburger and Hamilton, 1951*) (16–22 somites). Eggs were windowed and 0.2–0.5 mL of a 1:10 mixture of India ink (Fount India, Pelikan) and phosphate-buffered saline (PBS) was injected into the sub-blastodermal space. The window was lined with silicone grease, and the embryo raised to the level of the shell by pipetting PBS into the egg through the window, creating a bubble of PBS held in place by the grease. An incision was made between the neural tube and newly-formed sclerotome on one side of the embryo, and a single Affi-Gel Blue agarose gel bead (BioRad, Cibacron blue coupled to agarose, 30–50 µm diameter) impregnated with PACMA31 or PACMA56 (500 µM in PBS) was implanted into the resulting space (adjacent to the neural tube medially and notochord ventrally). Embryos were then re-incubated for 24–36 hr before fixing and processing for axon staining (see below) as whole-mounts or as (left and right) half-embryo-mounts. PACMA31 or PACMA56 solution was made by adding 0.5 ml dimethylsulfoxide to 2.2 mg PACMA to make a 10 mM stock solution. This was diluted x20 in PBS to make a 500 µM working solution, in which the Affigel beads were then placed for 2–3 hr at 21℃ prior to implantation.

## Preparation of tissue extracts

Stage 16–18 chick embryo trunks (comprising ectoderm, somites with DRG neurons, and motor axons, neural tube and notochord) or somite strips (ectoderm, somites with DRG neurons and motor axons) were dissected and immediately placed on solid $CO_2$ and transferred to −80℃ for longer storage. Trunks from ~60 embryos were homogenised in 1 ml solubilisation medium [2% w/v CHAPS in PBS made 1 mM with sodium orthovanadate and 1 tablet cØmplete protease inhibitor cocktail (Roche) per 50 ml of solution] on wet ice by shearing through a 20G then 26G needle. Further homogenisation was carried out with grinding resin (GE Healthcare) and electrically-driven disposable pestles (GE Healthcare). Following centrifugation at 14,000 g for 5 min at 4℃ to remove the grinding resin, the supernatant fluid was centrifuged at 100,000 g for 1 hr at 4℃ in a Beckman Optima TL ultracentrifuge using a TLS-55 rotor. Supernatant fluid was incorporated into liposomes as described by *Davies et al., 1990*. Pellets of 1718 cells were similarly extracted. Dissected rat (typically 3 months old) forebrain grey matter was stored at −80℃ and allowed to thaw on wet ice in the above solubilisation medium, ratio 0.5 g wet weight tissue to 2 ml medium. Following homogenisation in a Dounce Tissue Grinder (loose and tight fitting glass pestles were used in succession) and centrifugation at 14,000 g for 5 min at 4℃, the supernatant fluid was centrifuged at 100,000 g for 1 hr at 4℃ as described above. The clear supernatant fluid [14.9 ± 0.5 µg protein/µl (s.e.m.)] from the latter centrifugation was used for incorporation into liposomes.

## Growth cone collapse assays

These were carried out using whole- or half-DRGs dissected from embryonic day 7 (E7) chick embryos (stage 30–32) or (for retinal axons) from dissected pieces (~50 µM diameter) of E7 chick

retina. DRG explants were grown for 24 hr on glass coverslips coated with poly-L-lysine (Sigma-Aldrich) and laminin (Sigma-Aldrich), in the presence of nerve growth factor (NGF, 40 ng/ml, Sigma-Aldrich); full details of the assay method used in our laboratory have been published (*Cook et al., 2014*). Retinal explants were grown as for DRGs but without NGF and with medium supplemented with N-2 (Sigma-Aldrich, 100x concentrate) and bovine pituitary extract (200 µg/ml, Life Technologies). Cultures were fixed 1 hr after addition of each treatment, and growth cones were assessed by phase-contrast microscopy, observer-blind to treatment condition. Between 50 and >100 growth cones were assayed in each DRG culture. They were classified as spread or collapsed according to published morphological criteria (growth cones with two or fewer filopodia were scored as collapsed, see *Figure 3—figure supplement 1b*; *Cook et al., 2014*). We have previously shown the validity of this method using phase contrast-microscopy when compared to equivalent results using phalloidin staining (*Manns et al., 2012*). Results are presented as the mean percentage collapse of growth cones extended from individual DRG explants.

## Assessment of PDI inhibitors using collapse assay

Conditions of controls and reagents were assigned randomly to each culture and all experiments were blind-coded. Unless otherwise specified all components were added simultaneously and incubated for 1 hr ($37°C/5\%$ $CO_2$) before fixation. In experiments with antibody preparations, controls using the same concentration of non-immune rabbit IgG (Sigma-Aldrich G2018, Lot#051K7670), BSA and sodium azide were included. Immunodepletion experiments were performed using Magnetic Dyna Protein A beads (Invitrogen, 70 µl packed beads) washed twice in 0.1M phosphate buffer (pH 8) with 0.01% Tween 20 (1ml) by end-over-end mixing for 2.5 hr at 4°C. Washed beads were added to rat forebrain extracts (RFE) containing either anti-PDI (Sigma-Aldrich P7496 Lot#054K4801), non-immune rabbit IgG or bovine serum albumin (BSA) and mixed end-over-end for 1 hr at 4°C. Beads were removed on a magnetic separator and the extract subjected to a repeated extraction with fresh beads. Treated extracts were incorporated into liposomes.

## Live-staining of csPDI in whole-mounted somite-strips

Stage 22–24 chick embryos were removed from the egg and washed in Leibovitz's L-15 medium (Thermo Fisher Scientific) supplemented with 1% L-Glutamine-Penicillin-Streptomycin solution (Sigma-Aldrich). Embryos were pinned out along the A-P axis, ventral-up, in a Sylgard (Dow Corning) coated dish containing medium. After removal of the endoderm, embryos were re-pinned dorsal-up and the neural tube, intermediate mesoderm and lateral plate mesoderm were separated from the paraxial/somite mesoderm. Strips of somite mesoderm were dissected and collected in 4-chambered cell culture slides (BD Falcon) containing L-15 medium and sheep serum (Sigma-Aldrich, 10% v/v) as blocking solution, and slides incubated for 15 min. Primary anti-PDI antibody (Sigma-Aldrich P7496) or rhodamine-conjugated PNA (Vector labs) was added (1:500 v/v) to 3 chambers per slide and incubated for 1 hr at 38°C. Strips were fixed with 4% formaldehyde for 30 min, washed x3 with PBS, 5 min per wash, then incubated with secondary antibody (anti-rabbit IgG, Invitrogen) for 2 hr at 21°C. Controls for anti-PDI binding, each in the 4th chamber per slide, were: absence of primary antibody, primary antibody pre-absorbed with purified bovine PDI (Sigma-Aldrich, P3818, concentration 5x molarity of anti-PDI antibody), and rabbit IgG (1:500). Slides were mounted with Fluoromount G (SouthernBiotech) and viewed using a Zeiss Axioskop fluorescence microscope. Each staining procedure was repeated at least x3.

## Sclerotome and retinal cell staining and transfection

Dissected somite strips were collected in a 2 ml LoBind tube (Eppendorf) containing L15 medium, and sclerotome cells were dissociated with a 25G needle, after which 20 µl of cells were transferred into each chamber of a 4-chambered cell culture slide (BD Falcon) containing 490 µl medium per chamber pre-warmed at 37°C. To maintain sclerotome differentiation a notochord fragment was added to each well. Slides were cultured in a humidified box at 37°C for 16 hr, after which csPDI was assessed by anti-PDI- and PNA-staining as described above for somite strips. For retinal cells, eyes were removed from stage 22–24 embryos using a microscalpel, and retinal cells dissociated and stained as for sclerotome cells. For siRNA transfection, cells were incubated at 38°C for 16 hr. 10 µl of transfection mix [12.5 µg siRNA in 100 µl 5% glucose and 1.5 µl TurbofectTM (Thermo Fisher

Scientific)] in 490 µl of DMEM (Sigma-Aldrich) supplemented with B-27 (Life Technologies) and NGF (Sigma-Aldrich) was then added to cultures. After overnight incubation at 38 ˚C cells were washed x3 in DMEM and incubated for 3 hr with B-27/NGF-supplemented DMEM.

## Human astrocytic (1718) cell staining

Human astrocytoma-derived 1718 cells [CCF-STTG1 (ATCC CRL-1718TM)] were cultured in RPMI 1640 medium (ATCC modification, Gibco) supplemented with 10% fetal bovine serum (FBS; Gibco) and penicillin-streptomycin (Gibco). Cells were maintained at 37˚C in 5% $CO_2$. At each passage, cells were detached using Trypsin-EDTA (0.05%, Gibco), centrifuged at 1000 g for 5 min and plated in cell culture flasks (Nunc). After removing culture medium, cells were scraped in PBS or diethyl pyro-carbonate (DEPC) PBS and collected in an Eppendorf tube, then centrifuged at 2,000 g for 10 min at 4˚C. Re-suspended cells were washed once with RPMI 1640 and blocked with RPMI 1640/10% goat serum (Sigma-Aldrich) for 10 min at room temperature. They were plated in 4-well Millicell EZ slides (Millipore) coated with poly-L-lysine (0.01%, Sigma-Aldrich), at a concentration of 50,000 cells per well (1.7cm2). Cells were incubated with primary antibody in RPMI 1640/1% goat serum for 30 min at 4˚C, washed x3 with RPMI 1640, then incubated with secondary antibody in RPMI 1640/1% goat serum for 1 hr at 4˚C and washed x3 with RPMI. Cells were fixed with 4% w/v formaldehyde and 15% w/v sucrose in PBS, pH 7.4, for 10 min at 21˚C, then washed for 5 min x3 with PBS. For live-cell imaging of the ER, cells were washed once with HBSS and incubated with ER-TrackerTM Green (BODIPY FL Glibenclamide, Life Technologies) for 20 min at 37˚C. Cells were then washed x3 with HBSS and slides were mounted using Fluoromount-G (Southern Biotech). For intracellular immunos-taining cells were washed x1 with PBS, fixed with 4% w/v formaldehyde/15% w/v sucrose in PBS for 10 min at 21˚C, then washed for 5 min x3 with PBS. Cells were blocked with PBS with or without 0.1% Triton X-100 (PBS-T) and 10% v/v goat serum for 1 hr at 21˚C. They were then incubated in pri-mary antibody in PBS-T/1% goat serum overnight at 4˚C, washed x3 with PBS, incubated for 1 hr in secondary antibody in PBS-T at room temperature, then washed x3 with PBS. Nuclear staining was performed with 4',6-diamidino-2-phenylindole (DAPI, Sigma-Aldrich) diluted 1:4000 in PBS, or with Hoechst diluted 1:5000 in PBS. Primary antibodies were: rabbit anti-PDI (IgG polyclonal, Sigma-Aldrich) used at 1:250 (live-staining) or 1:500 (post-fixation staining); anti-calnexin (Abcam, clone 6F12BE10, mouse IgG2b) used at 1:100X (live-staining) and 1:200 (post-fixation staining). Secondary antibodies were goat anti-rabbit IgG and goat anti-mouse IgG (Alexa Fluor 594) used at 1:500.

## Embryo fixation and dehydration

Stage 19–22 embryos were washed x2 in PBS before removing the extra-embryonic tissues. Embryos were fixed in 4% w/v formaldehyde for 2 hr at 21˚C, or overnight at 4˚C, then rinsed in PBS on a mechanical shaker for 5 min and dehydrated through a series of 10 min washes x1 with 25, 50, 75% v/v methanol/PBS and 100% methanol. After one further 30 min wash in 100% methanol, embryos were stored in methanol at −20˚C until required.

## Axon staining

After rehydration into PBS-T, embryos were blocked in PBS-T/10% goat serum for 3 hr at 21˚C, then incubated in fluorescein-conjugated PNA (Vector Labs) for sclerotome csPDI, or in anti-tubulin βIII (clone TUJ1, Mouse IgG2a, BioLegend) for axon staining, both at 1:500 in PBS-T/10% v/v goat serum for 12–18 hr at 4˚C. Embryos were then washed x4 for 20 min with PBS-T. Secondary antibody (per-oxidase goat anti-mouse IgG, Jackson ImmunoResearch) was used at 1:500 in PBS-T/10% goat serum for 2 hr at 21˚C, followed by 20 min washes x4 in PBS-T. Embryos were then incubated with 500 µg/ml diaminobenzidine (DAB) substrate and 0.006% $H_2O_2$ in PBS/0.5% Triton, and the colour reaction was developed for 5–10 min at 21˚C.

## Vibratome sectioning

Formaldehyde-fixed embryos were embedded in 10% gelatin (bloom 300, Sigma-Aldrich) in PBS at 38˚C for 30 min. Cryomolds (Tissue-Tek) with gelatin were set at 21˚C for 15 min, after which embryos were transferred to them and the gelatin flattened and set at 4˚C for 30 min. Blocks were cut and fixed with 4% formaldehyde at 4˚C for at least 72 hr, then washed for 10 min x3 in PBS,

trimmed and mounted in a Leica VT1000 S vibratome. Sections were cut at 70 µM using a steel blade and mounted on glass slides (VWR International) using Fluoromount G.

## Primary cultures of rat cortical astrocytes

Cortical hemispheres from neonatal rat pups (P1-P3) were isolated and dissected in ice-cold DMEM containing penicillin-streptomycin (Gibco). Care was taken to remove meninges and white matter. Cortices from up to 12 pups were pooled and sub-divided in a Petri dish using a razor blade. The tissue was transferred to a 15 ml Falcon tube and spun briefly, then resuspended in 2 ml papain solution [0.75% v/v of papain (25 mg/ml, 17 U/mg protein, Sigma-Aldrich), 40 µg/ml DNase I type IV, 2 mM L-Cysteine in DMEM with penicillin-streptomycin] and incubated for 1 hr at 37°C with occasional resuspension. The enzymatic digestion was quenched by adding 2 ml trypsin-inhibitor solution [500 ug/ml BSA, 40 µg/ml DNase I type IV, 1 mg/ml Trypsin inhibitor (Sigma-Aldrich)]. Cells were then dissociated by mechanical resuspension in 1 ml ovomucoid solution and collected by centrifugation in a 10 ml trypsin-inhibitor solution. They were resuspended in culture medium and plated in poly-D-lysine-coated flasks (cells from 1 to 1.5 brains in one 75 cm$^2$ culture flask). Cells were cultured at 37°C/5% CO2 in DMEM (Gibco) supplemented with 10% FBS (Gibco) and penicillin-streptomycin. After 7-10d culture cells were shaken in an orbital shaker at 350–400 rpm (1.9 cm orbital radius, MaxQ 4450, ThermoFisher Scientific) at 37°C to obtain a culture of cortical astrocytes. Microglia, neurons and oligodendrocytes were detached after an overnight shaking, and medium was then replaced. Cultures consisted in >90% GFAP-positive cells.

## siRNA knockdown of csPDI

The fluorescein-labelled siRNA used to knock down csPDI in the chick embryo was designed according to the sequence of an antisense phosphorothioate (S-oligo; nuclease-resistant oligonucleotide) successfully used by *Zai et al., 1999* to knock down csPDI/PDIA1/P4HB in a human erythroleukemia (HEL) cell line. These authors designed three antisense S-oligos against human PDIA1/P4HB mRNA, and one of these reduced the cell surface expression of P4HB significantly (by 74 ± 9.2% compared to the scrambled S-oligo control). The sequence for the successful oligo was 5′-GCAGCGAGAC TCCGAACACGGTA-3′, found in the 3′ UTR of the human PDIA1/P4HB mRNA. This sequence was used to find an appropriate target sequence in the 3′ UTR of chicken PDIA1/P4HB (see Key Resources Table), and was selected using BLAST NCBI anSfold (Wadsworth Center) to ensure maximum binding. A FITC-labelled control scrambled siRNA was designed using InvivoGen siRNA Wizard software http://www.sirnawizard.com/scrambled.php (see Key Resources Table). All sequences were subjected to NCBI BLAST to ensure gene-specificity and to avoid mis-targeting. Rescue experiments used a plasmid encoding a fusion protein of mature human PDIA1 (18–508), tagged at its N-terminus with a bovine pre-pro-trypsinogen signal peptide (bPPTSP) and a FLAG-M1 epitope that is exposed after cleavage of the signal peptide (kind gift of Prof David Ron, Department of Clinical Biochemistry, University of Cambridge) (*Zito et al., 2010*).

## Primer design

Transcript sequences for selected genes were obtained via the National Center for Biotechnology Information (NCBI) GenBank and Ensembl. Primer pairs for each transcript were designed using the Primer-Blast tool available from the NCBI (http://www.ncbi.nlm.nih.gov/tools/primer-blast/). Primers were selected according to the following rules: (i) primer length 17–30 base pairs; (ii) CG content 50–60%; (iii) melting temperature 55–80°C; (iv) resulting amplification product 400–1200 base pairs. All potential primers were checked against the G. gallus (taxid:9031) genomic database using the Basic Local Alignment Search Tool (BLAST) from NCBI. The outputs from this last step were used to exclude all primers giving more than one significant region of identity (80% cut-off) against the whole chicken genome, or sharing more than 70% similarity with other genes. Selected primers were synthesized (Sigma-Aldrich) with the T7 promoter primer sequence (TAATACGACTCACTATAGG-GAG) appended to the 5′ end of the reverse primer, to allow direct generation of digoxigenin-labelled antisense RNA probe by in vitro transcription using T7 RNA polymerase.

Polymerase chain reaction (PCR) to prepare template for riboprobe synthesis cDNA samples (2 µl) were pipetted into a 200 µl thin-wall centrifuge tube and 36 µl of DEPC-treated water, 6 µl of primer and 50 µl of Reddy Mix PCR Master Mix (AB Gene) was added to each. The contents of the

tube were briefly mixed and spun down. Tubes were then placed on a heating block of a hot-lid thermal cycler pre-heated to 95℃. Cycling commenced with an initial 2 min denaturation step at 95℃ followed by 34 cycles of 95℃ for 25 s, annealing at 50℃ for 45 s and elongation at 72℃ for 1 min; cycling finished with an extension step of 72℃ for 5 min. The PCR product length was checked by agarose gel electrophoresis, and the products stored at −20℃ until needed for riboprobe synthesis.

## Riboprobe synthesis

20 µl in vitro transcription reactions were prepared by adding to a 200 µl thin-wall PCR tube in the following order: 9 µl DEPC-treated water, 4 µl nucleoside triphosphate (NTP) mix (2.5 mM ATP, 2.5 mM CTP, 2.5 mM GTP, 1.67 mM UTP, 0.833 mM digoxigenin-11-UTP), 2 µl T7 transcription buffer (Ambion), 2 µl T7 RNA polymerase, 1 µl RNase inhibitor (Invitrogen), and 2 µl PCR product. The tube contents were mixed by pipetting and briefly spun in a microfuge ($\leq$1000 g) to settle them. The tube was further incubated at 37℃ in a thermal cycler for 2 hr, after which 1 µl DNase I was added and the tube further incubated in a thermal cycler for 15 min at 37℃. To stop the reaction, 1 µl of 0.5M EDTA was added and mixed by pipetting, and the tube contents spun down. The probe was then analysed using a Picodrop spectrophotometer.

## Isolation of RNA for antisense RNA probes and cDNA synthesis

Embryos were rinsed with cold diethyl pyrocarbonate (DEPC)-PBS and transferred to a methanol-washed Petri dish coated with Sylgard. The extra-embryonic membranes were removed using Watchmaker's forceps pre-cleaned with RNAse Zap (Ambion). Embryos were placed in RNAlater (Ambion) and stored overnight at 4℃. Total RNA was extracted using silica-membrane RNeasy spin columns (Qiagen) according to manufacturer's instructions. cDNA was synthesized using the iScript cDNA synthesis kit (Bio-Rad) according to manufacturer's instructions and stored at −20℃.

## Whole-mount in situ hybridization (WMISH)

Our procedure was based on *Wilkinson, 1998*. Embryos were rehydrated into PBS-T through a series of 75% v/v methanol/ultra-pure water, 50% v/v methanol/ultra-pure water, 25% v/v methanol/PBST. Embryos were transferred into 18-well plates (Nunc). Unless otherwise specified, all reagents were diluted in PBS-T and washes were for 10 min in PBS-T on a rocking platform at 21℃. To increase probe permeability embryos were incubated at 21℃ in 10 µg/ml proteinase K (Roche) for the following durations: embryos to stage 15 for 5 min, stage 16–18 for 10 min and stage 19–24 for 15 min. Embryos were rinsed x1, post-fixed for 20 min in 4% formaldehyde and washed x2 to remove fixative. They were equilibrated with hybridization mix [50% v/v formamide, 5X SSC (Sigma-Aldrich), 2% blocking powder (Boehringer, 1096176), 0.1% Triton X-100, 0.1% CHAPS (Sigma-Aldrich), 1 mg/ml tRNA (Sigma-Aldrich), 5 mM EDTA, 50 µg/ml heparin] by rinsing x1 in a 1:1 mixture of PBST/hybridization mix and then x2 in hybridization mix. Plates were then placed at 67℃ in a hybridization rocking oven for a minimum pre-hybridization of 2 hr to 12 hr maximum, after which the solution was changed to pre-warmed hybridization solution containing 1 µg/ml RNA probe, and incubated for at least 12 hr to 72 hr maximum. In order to avoid cross contamination, WMISH probes were well separated when the hybridization was being done; vials were leak-proof, and each probe was used no more than x3. After incubation embryos were rinsed x2 and washed x1 with pre-warmed hybridization mix, washed x2 for 30 min with hybridization mix, and then x2 with a 1:1 mixture of hybridization mix/PBS-T at 60℃. Embryos were then rinsed x3 with PBS-T. Hybridization solution was eliminated by 30 min washes x7 in PBST at 21℃ in a rocking shaker. To block non-specific binding, embryos were incubated for 1–3 hr in blocking solution (10% v/v Sigma-Aldrich sheep serum in PBS-T) at 21℃. This was replaced with blocking solution containing alkaline phosphatase-conjugated anti-digoxigenin Fab fragments (Roche) at 1:2000 dilution, and embryos were incubated further for 12–18 hr at 4℃. The antibody was removed by rinsing the embryos x3 in PBST with 1 mM levamisol, followed by 4 hr of washes with buffer changes every 30 min; in some cases embryos were left overnight at 4℃. Alkaline phosphatase was detected using a mixture of 4-nitro blue tetrazolium chloride (NBT) and 5-bromo-4-chloro-3'-indolyphosphate (BCIP). Embryos were first washed x2 in NTMT (100 mM NaCl, 100 mM Tris-HCl pH 9.5, 50 mM $MgCl_2$, 0.1% Triton X-100), followed by addition of the reaction mixture (4.5 µl/ml NBT and 3.5 µl/ml BCIP in NTMT). Reactions were left in the dark until a deep purple colour had developed; this could take 3 hr to 5d, and in the latter

case the stain solution was replaced daily. Embryos were then washed x3 in PBS-T and fixed in 4% w/v formaldehyde for 12–18 hr at 4°C. The fixative was removed by several PBS washes. Embryos were imaged using a Leica dissecting microscope and prepared for vibratome sectioning.

## siRNA preparation

Lyophilized FITC-labelled RNA duplexes (Dharmacon Thermo Scientific) were obtained in 2' deprotected, annealed and desalted form, dissolved in PCR grade water (Roche) at 3 µg/µl and stored in aliquots at −80°C. The transfection solution was 1 µg/µl siRNA, 10% polyethylene glycol (PEG) (Carbowax 6000, Union Carbide) and 20% TurbofectTM (Thermo Fisher Scientific, Catalog # R0541). For 2.8 µl of siRNA preparation, 1 µl of siRNA, 1.2 µl of 20% PEG stock and 0.6 µl of TurbofectTM were incubated at 21°C for 30 min before application. This technique was also tested using pCAβ-EGFPm5-mU6 (Bron et al., 2007), a kind gift of Dr. Matthieu Vermeren (Department of Physiology, Development and Neuroscience, University of Cambridge, UK). The final transfection solution contained 2 µg/µl plasmid, 10% PEG and 40% TurbofectTM. For a final 5 µl of solution, 1 µl of plasmid, 2 µl of 20% PEG stock and 2 µl of TurbofectTM were used; this solution was only used once. For siRNA delivery in ovo, borosilicate glass capillaries (WPI, outside diameter 1.5 mm, inside diameter 1.12 mm) were pulled on a Narishige Puller PC-10 at 62°C. Tips were broken to obtain a suitably narrow internal diameter and capillaries attached to a rubber tube/mouth-pipette.

## In ovo transfection

Eggs were cleaned with methanol and 3–4 ml of ovalbumin removed using a 19G needle and syringe. The upper side of the egg was reinforced with adhesive tape and a window ~1 cm diameter cut through shell and tape using curved scissors. The embryo was raised to the level of the window by re-pipetting the ovalbumin, and visualized by injection into the yolk of ~0.2 ml black ink (Pelikan Fount India, 5% in PBS). A small incision was made in the vitelline membrane overlying the posterior part of the embryo using a microscalpel. The glass capillary containing siRNA/plasmid transfection solution was inserted into the most posterior and newly formed somite of stage 10–14 embryos, and carefully advanced anteriorly within or immediately ventral to the somite mesoderm on one side of the embryo, parallel to the neural tube and dorsal to the endoderm, until the most anterior accessible somite was reached. The capillary was then slowly withdrawn and siRNA was injected into 8–12 successive sclerotomes, each over 5–10 s (~0.05 µl total volume injected per embryo). Care was taken to avoid the upper two cervical segments where the avian spinal accessory nerve exits and ascends immediately adjacent to the neural tube. After pipette withdrawal the embryo was returned to the egg by removing 5 ml of ovalbumin, and the window closed with adhesive tape. Each egg was re-incubated for 24 hr, when siRNA delivery in somites on the injected side was confirmed by the presence of fluorescence in >8 consecutive somites in ovo viewed by epifluorescence microscopy. Eggs were then incubated for 24 hr further to stages 22–24, when embryos were processed for somite strip or sclerotome cell culture and staining, or immunohistochemistry or in situ hybridization, all as described above.

## Antibodies

Polyclonal rabbit anti-PDI (Sigma-Aldrich P7496) was prepared using PDI purified from bovine liver as immunogen. The whole serum was fractionated and further purified by ion-exchange chromatography to provide the IgG fraction essentially free of other rabbit serum proteins. In this study Lot#054K4801 (protein content 7.1 mg/ml in 0.1M phosphate buffered saline pH 7.4 containing 15 mM sodium azide) was used. Polyclonal anti-PDI antibody Abcam ab31811 (0.4 mg/ml PDI Ab, 1% BSA, 2% Sodium Azide) was raised in rabbit against a synthetic peptide corresponding to human PDI amino acids 400–500 conjugated to keyhole limpet haemocyanin and immunogen affinity-purified; this contained IgG at 0.4 mg/ml in 1% BSA, and PBS pH 7.4 containing 0.02% sodium azide as preservative.

## PDI inhibitors

Bacitracin (Fluka Lot#13Z3372) was examined for protease activity using azocasein (Sigma-Aldrich Lot#039K7002) as a substrate and protease from Bacillis liceniformis (Sigma-Aldrich Lot#040M1970V) as a standard (Rogelj et al., 2000). A trace (<0.05%) of enzyme was detected and enzyme-free

bacitracin reagent was prepared by gel filtration through Sephadex G100 (*Rogelj et al., 2000*). 16F16 (Lot#051M4613V), phenylarsine oxide (Lot#056K1654) and Rutin hydrate (quercetin-3-rutinoside:≥94%[HPLC] Lot#BCBH6323V) were purchased from Sigma-Aldrich. T3 (3,3′,5′ triiodo-L-thyonine: Sigma-Aldrich ≥ 95%[HPLC] Lot#016K1628V) was acetylated with acetic acid N-hydroxy-succinimide ester (Apollo Scientific) as described (*Gallina et al., 2002*) and the product shown to be homogeneous by thin layer chromatography. The propynoic acid carbamoyl methyl amines PACMA31 and PACMA56 were synthesised as described (*Xu et al., 2012*).

## Other reagents

S-Nitrosoglutathione (Lot#055M403V), L-glutathione reduced (G4251 Lot#SLBH7927V), L-glutathione oxidised (G4626 Lot#100K727625), DL-dithiothreitol (43819, Lot#BCBG3415V) and eosin 5-isothiocyanate (Lot#BCBK9368V) were obtained from Sigma-Aldrich, and L-homocysteine (Lot#B1612) from Santa Cruz Biotechnology. Sema 3A/Fc chimera was from R and D Systems (Lot#1250–53). Agarose bound Peanut Agglutinin (Lot#ZA0611; binding capacity >4.5 mg asialofetuin/ml of gel) and Agarose bound Jacalin (Lot#ZA1021) were from Vector Laboratories. Cyanogen bromide-activated Sepharose 4B beads (Sigma C9142) were used to couple purified bovine serum albumin (BSA). After coupling the gel was blocked with 1 mM ethanolamine. Beads used in these experiments contained 9.48 mg BSA per ml of settled gel.

## Protein assay

Protein assays were performed with bicinchoninic acid reagent [Pierce BCA protein assay kit (Lot#QA214075); Sigma Bicinchoninic acid solution (Lot#SHBH4613V) and copper(II) sulphate (Lot#SLBJ6167V) with bovine serum albumin (Pierce Lot#BB42996, 2.0 mg/ml in 0.9% NaCl)] as standard, and using the enhanced protocol (60°C for 30 min). A separate standard curve was constructed for each assay and the sample was subject to at least 3 separate dilutions which were each determined in duplicate.

## Purification of csPDI from somites

A total of 400 chick embryo trunks were fractionated by affinity chromatography on agarose-bound-PNA (Vector Labs), following procedures used previously in the laboratory (*Davies et al., 1990*). Care was taken to elute the affinity column with 0.5M NaCl 1% CHAPS (w/v) and 100 mM Tris-HCl (pH7.5), followed by elution with 0.4M lactose/2% CHAPS (w/v) in PBS. Eluates (20 µL) were concentrated using StrataClean Resin (Agilent Technologies) (*Bonn et al., 2014*; *Otto et al., 2017*). Protein bound to the resin was eluted using SDS reducing sample buffer with heating for five minutes at 95°C, followed by centrifugation (10000 g for 1 min). The supernatant containing the proteins was fractionated on slab gels (7.5% acrylamide separating gel; 5% stacking gel). Samples were examined under reducing conditions and electrophoresis was performed in 25 mM Tris (pH 8.3), 192 mM glycine, 0.1% SDS. Molecular weight markers (BenchMark Protein Ladder, Invitrogen) were also run. The gel was developed with MS-compatible silver stain using the protocol of *Blum et al., 1987*. The band was excised in a laminar flow hood and submitted for mass spectrometry analysis (Alta Bioscience, UK).

## Identifying somite proteins that act as a substrate for S-nitrosylation

The Pierce S-Nitrosylation Western Blot Kit (ThermoFisher Scientific) was used, in which a lower background is obtained with iodoTMTzero reagent (Lot# PA19543) compared with biotin labelling. A cell free assay was prepared in which 650 somite strips were homogenized in HENS buffer [1 ml + 10 µl protease inhibitor cocktail (Sigma Lot# 033M4023V)] using an electrically-driven disposable pestle and grinding resin (GE Healthcare). Following centrifugation at 1000 g for 1 min at 10°C to remove the resin, the homogenate was centrifuged at 10,000 g for 20 min. Aliquots of homogenate containing 200 µg protein in 200 µl of HENS buffer made 200 µM with GSNO. Reduced glutathione was used as a negative control. After incubation at room temperature in the dark for 45 min, unreacted GSNO was removed using P6 microcolumns (BioRad) and the samples blocked with methyl methanethiosulfonate. Labelling with iodoTMT reagent was performed in the presence of sodium ascorbate and controls in the presence of water. Protein samples (48 µg) were fractionated on NuPAGE 4–12% Bis Tris gels in MOPS buffer and blotted onto Hybond C-extra nitrocellulose

membrane using NuPAGE transfer buffer (Thermo Fisher Scientific) containing antioxidant. Blots were probed with anti-TMT antibody (1:1000, Lot#OH190916) purified from mouse ascites fluid with Pierce Goat Anti-Mouse IgG (H+L) HRP conjugate (Lot#OI192080).

### Identification of LC1 in somite extracts

An extract of stage 19/20 chick embryo trunks was prepared in HENS buffer containing protease inhibitor (as above for somite strips) and the protein content quantified. An aliquot containing 25 µg protein was fractionated on a 4–12% Bis-Tris gel and blotted onto Hybond C-extra nitrocellulose membrane using NuPAGE transfer buffer (Thermo Fisher Scientific) containing antioxidant. The blot was cut in half just above the 41K marker and the top half of the membrane was probed with rabbit anti-tubulin (Sigma, Lot#50K4813) followed by goat anti-rabbit IgG (HRP, Abcam Lot#GR3231028-7, 1:20,000). The bottom half was probed with MAP-1B (LC1) mouse monoclonal antibody against amino acids 2257–2357 of MAP-1B of mouse origin (Santa Cruz Biotechnology, Inc, sc-136472) followed by goat anti-mouse IgG (HRP, Pierce Lot#TE262980, 1:20,000). Blots were blocked in 5% non-fat dried milk (BioRad) in TBST and thoroughly washed x5, each for 5 min, in TBST. In both above experiments blots were treated with Millipore Immobilon Western Reagent (Lot#1710401) and exposed to film.

### Action of D-NAME and L-NAME on S-nitrosylation of LC1

Somite extract (200 µg protein for each condition) was incubated at 37°C for 1 hr in the presence of 300 µM D-NAME (Sigma, Lot#BCBM7105V) or L-NAME (Sigma, Lot#BCBT1028). A control experiment with somite extract and buffer alone was included. Subsequently extracts were made 200 µM in S-nitrosoglutathione and left at room temperature for 45 min before being processed as above and subjected to fractionation on a Nu-PAGE 4–12% Bis-Tris gel in MOPS buffer followed by blotting on Hybond C-extra. Care was taken to load equal amounts of protein (15 µg per lane). Processed samples were assayed for protein levels with a Qubit Fluorometer 2.0 (Thermo Fisher Scientific) using the Qubit protein assay (quantitation range 0.25–5 µg) to achieve the same quantity of sample in each lane. The blot was cut in half below the 53K molecular weight marker and the top half probed with rabbit anti-tubulin (Sigma, Lot#50K4813, 1:20,000) and goat anti-rabbit IgG (HRP, Abcam Lot#GR3231028-7) followed by Millipore Immobilon Western Reagent. The damp membrane was examined using an iBright FL1500 imaging system (Thermo Fisher Scientific) and the digital image caught directly by the instrument to authenticate that somite extract was loaded in every lane. The bottom half of the blot was probed with anti-iodoTMT (Lot#PH204668, 1:1000) and goat anti-mouse IgG (H+L) HRP, 1:20,000, followed by Clarity Western ECL substrate (mid-femtogram-level sensitivity, BioRad). The damp membrane was examined in the iBright FL1500 imaging system and the digital image captured.

### Western blot of rat cortical astrocyte cell surface proteins

Two month-old wild-type rats (Rattus norvegicus) were used as a source of neonatal rat cerebral cortical astrocytes. Four flasks of cortical astrocytes (in DMEM with 10% FBS and penicillin/streptomycin) at 95% confluence were subjected to biotinylation using a commercial 'Cell Surface Protein Isolation Kit' (ThermoScientific, Prod#89881, Lot#RD234938). Following labelling of the cell surface proteins with EZ-link-Sulfo-NHS-SS-Biotin reagent, the biotinylated proteins were captured on NeutrAvidin resin, washed thoroughly and the bound proteins released by cleavage of the S-S bond by treatment with freshly prepared SDS-PAGE sample buffer made 50 mM with respect to DTT. One-third of this eluate was fractionated by SDS-PAGE on a 7.5% polyacrylamide resolving gel (120 × 80mmx3mm; 5% stacking gel) and blotted onto Hybond C-extra nitrocellulose membrane (Amersham Biosciences Batch No. 319063) using 25 mM Tris (pH8.3), 192 mM glycine and 0.1% SDS electrophoresis buffer. The blot was blocked with 5% Blotting Grade Non Fat Dry Milk (BioRad) and developed with 1:20,000 anti-PDI (Sigma P7496) followed by 1:2000 Tidy Blot Western Blot Detection Reagent-HRP (BioRad, Batch#160129) and the use of Immobilon Western Chemiluminescent HRP substrate (Millipore).

## Assessment of reductase activity in purified PDI

Di-E-GSSG was prepared by the reaction of eosin isothiocyanate (Sigma-Aldrich) with L-glutathione oxidised (Sigma-Aldrich Lot#100K72765) as described in detail by *Raturi and Mutus, 2007*. Four samples of PDI (4 µg) were incubated in 20 µl 100 mM potassium phosphate (pH7), made 1.5 µM with respect to calcium and magnesium chloride, with 20 µl packed PNA-agarose beads (Vector lot ZC0504) with a capacity to bind >90 µg asialofetuin to 20 µl beads. The beads were kept at 5°C over 18 hr with intermittent mixing. Following centrifugation at 14,000 g for 5 min at 4°C and a further wash with 30 µl of buffer, the combined supernatant fluids were added to the reaction mixture to give a maximum concentration 128 nM PDI. Reductase activity was monitored as above (*Raturi and Mutus, 2007*). Fluorescence was measured in a Biotronix Fluorometer (Electronics and Instrumentation Services for Biological Science, University of Cambridge).

## Statistics

A non-parametric Kruskal-Wallis one-way ANOVA was used for comparison of data sets. The Mann-Whitney U test was used for comparison between treatment conditions in collapse assays. For comparison between three or more data points a two-way ANOVA was performed, followed by a post-hoc Bonferroni correction. No statistical methods were used to predetermine sample size. Graphs and figures were produced with GraphPad Prism 7.0 and Adobe Photoshop CS6. Histograms show mean +/- s.e.m.; see *Supplementary file 1* for statistical data.

## Acknowledgements

We thank C Stern, M Bate, C Holt, RB Heap and O Paulsen for comments on the manuscript. The initial parts of this work were supported by grants from the Medical Research Council, the Wellcome Trust, and the Howard Hughes Medical Institute, and more recently by grants from the Rosetrees Trust and Trinity College, Cambridge. JS was also supported by a studentship from the International Spinal Research Trust; EW was in receipt of an Undergraduate Summer Vacation Research Scholarship from the Anatomical Society; GR-V was in receipt of an Amgen Foundation Summer Research Scholarship.

## Additional information

### Funding

| Funder | Grant reference number | Author |
| --- | --- | --- |
| Medical Research Council | | Geoffrey MW Cook<br>Roger J Keynes |
| Wellcome | | Geoffrey MW Cook<br>Roger J Keynes |
| Spinal Research | | Julia Schaeffer |
| Trinity College, University of Cambridge | | Roger J Keynes |
| University of Cambridge | | Geoffrey MW Cook<br>Catia Sousa<br>Julia Schaeffer<br>Katherine Wiles<br>Prem Jareonsettasin<br>Asanish Kalyanasundaram<br>Eleanor Walder<br>Catharina Casper<br>Serena Patel<br>Pei Wei Chua<br>Gioia Riboni-Verri<br>Mansoor Raza<br>Nol Swaddiwudhipong<br>Andrew Hui<br>Ameer Abdullah<br>Saj Wajed<br>Roger J Keynes |

| Rosetrees Trust | | Geoffrey MW Cook<br>Julia Schaeffer<br>Roger J Keynes |
|---|---|---|
| The Anatomical Society | | Eleanor Walder |
| Amgen Foundation | Summer Scholarship | Gioia Riboni-Verri |

The authors declare that the funders provided research equipment and laboratory consumables, as well as salary support for Julia Schaeffer, Eleanor Walder and Gioia Riboni-Verri.

### Author contributions

Geoffrey MW Cook, Conceptualization, Resources, Data curation, Formal analysis, Supervision, Funding acquisition, Validation, Investigation, Visualization, Methodology, Writing - original draft, Project administration, Writing - review and editing; Catia Sousa, Conceptualization, Investigation, Methodology, Writing - review and editing; Julia Schaeffer, Conceptualization, Data curation, Supervision, Funding acquisition, Investigation, Methodology, Writing - review and editing; Katherine Wiles, Prem Jareonsettasin, Conceptualization, Data curation, Formal analysis, Investigation, Writing - review and editing; Asanish Kalyanasundaram, Catharina Casper, Pei Wei Chua, Gioia Riboni-Verri, Nol Swaddiwudhipong, Ameer Abdullah, Saj Wajed, Data curation, Investigation, Writing - review and editing; Eleanor Walder, Serena Patel, Data curation, Formal analysis, Investigation, Writing - review and editing; Mansoor Raza, Conceptualization, Data curation, Software, Formal analysis, Validation, Writing - review and editing; Andrew Hui, Data curation, Investigation; Roger J Keynes, Conceptualization, Resources, Data curation, Formal analysis, Supervision, Funding acquisition, Validation, Investigation, Methodology, Writing - original draft, Project administration, Writing - review and editing

### Author ORCIDs

Roger J Keynes ⓘ https://orcid.org/0000-0002-1557-7684

### Ethics

Animal experimentation: Chick embryos were used for this work, and all experiments were carried out at earlier developmental stages than those that require ethical approval.

### Decision letter and Author response

Decision letter https://doi.org/10.7554/eLife.54612.sa1
Author response https://doi.org/10.7554/eLife.54612.sa2

## Additional files

### Supplementary files

• Supplementary file 1. Statistical data for *Figure 1—figure supplement 1b–f*; *Figure 2b–g*; *Figure 2—figure supplement 1a–g,i,l,n,o*; *Figure 3a–h*; *Figure 3—figure supplement 1a–k*; *Figure 4a–e*; *Figure 4—figure supplement 1a–f,h*.

• Transparent reporting form

### Data availability

All data generated or analysed during this study are included in the manuscript and supporting files.

The following previously published dataset was used:

| Author(s) | Year | Dataset title | Dataset URL | Database and Identifier |
|---|---|---|---|---|
| Parkkonen T, Kivirikko KZ, Pihlajaniemi T | 1988 | Molecular cloning of a multifunctional chicken protein acting as the prolyl 4-hydroxylase beta-subunit, protein disulphide- | https://www.uniprot.org/uniprot/P09102 | UniProtKB, P09102 |

isomerase and a cellular thyroid-hormone-binding protein. Comparison of cDNA-deduced amino acid sequences with those in other species.

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
