## [Decision Letter]

**Acceptance summary:**

This excellent study reveals a mechanism for how growth of sensory and motor axons is restricted to the anterior half of each somite-derived sclerotome in the spinal cord, during generation of the segmented patterning of nerves during early peripheral nervous system development. The authors show that the cell surface protein disulphide isomerase contributes to such axonal segmentation by serving as an impenetrable barrier for growth cones in the posterior somite through acting on a nitric oxide/S-nitrosylation-dependent signal transduction pathway regulating the growth cone cytoskeleton, inducing axon growth cones to traverse the anterior half of each somite as they extend towards their body targets.

**Decision letter after peer review:**

Thank you for submitting your article "Regulation of nerve growth and patterning by cell surface protein disulphide isomerase" for consideration by *eLife*. Your article has been reviewed by three peer reviewers, and the evaluation has been overseen by a Reviewing Editor and Marianne Bronner as the Senior Editor. The following individual involved in review of your submission has agreed to reveal their identity: Ann Rajnicek (Reviewer #2).

The reviewers have discussed the reviews with one another and the Reviewing Editor has drafted this decision to help you prepare a revised submission.

Summary:

Your study on how the growth of sensory and motor axons are restricted to the anterior half of each somite-derived sclerotome, generating the segmented patterning of the early PNS, was viewed favorably by the reviewers. The demonstration of a key role for cell surface protein disulphide isomerase (PDI) via an NO-based mechanism in the establishment of such segmental axon patterning was deemed a major and important contribution. To ensure that the manuscript is accessible to the wide audience of *eLife* and to be clear even to developmental biologists, the reviewers and I ask that you insert a diagram in the first figure that would depict the somite arrangements and axon tracts, to set out the hypothesis, then a summary diagram in a final figure that would pull together the various strands of the results. Also, the reviewers call for examples of growth cone collapse and time course images, and for minor text modifications. I have left the reviews intact so that you can read the many positive comments on your study.

Essential revisions:

All of the revisions requested below are essential, and none call for performing new experiments.

Reviewer #1:

The restriction of the growth of sensory and motor axons to the anterior half of each somite-derived sclerotome, generating the segmented patterning of the early PNS, and the demonstration that this is due to contact repulsion exerted by the posterior sclerotome cells is one of the first and classic examples of this fundamental axon guidance mechanism. However, the molecular basis of segmental patterning is not understood. Here Cook and colleagues provide evidence for a key role for cell surface protein disulphide isomerase (PDI) via an NO-based mechanism in the establishment of segmental axon patterning and as such make a major and important contribution to our understanding.

Because they have previously shown that the lectin peanut agglutinin (PNA) binds to the posterior half sclerotome cells and that immobilised PNA depletes the collapse activity of the posterior sclerotome which is recovered by lactose elution, they subjected a major band in silver-stained SDS PAGE preparations of lactose elutes to tryptic digestion and mass spectrometry. This revealed numerous peptides distributed throughout PDIA1/P4HB, a member of the PDI family. They additionally showed that PDI staining appears in the posterior sclerotome shortly before the segmental emergence of sensory and motor axons. Importantly, they showed that in ovo microinjection of PDI siRNA, but not a scrambled control, disrupted the segmental patterning of early axons. As an alternative demonstration of the role of PDI, they microinjected a molecule that has been shown to specifically disrupt one of the active sites of PDI. Like PDI siRNA, this too disrupted segmental axon growth. To elucidate how PDI causes growth cone collapse, they investigated the potential involvement of an NO, based on the previous demonstration that NO causes growth cone collapse and that PDS catalyses NO entry by a transnitrosation mechanism. They showed in a DRG growth cone collapse assay that PDI (but not inhibited PDI) promotes NO donor-induced collapse. Moreover, somite extract-induced collapse was prevented by pre-treating DRG with an NOS inhibitor. Because S-nitrosylation of the MAPB1 light chain subunit LC1 has been shown to mediates calcium ionophore-induced growth cone collapse, they investigated whether a similar mechanism occurs in somite-induced growth cone collapse. Western blotting revealed that only one band, that corresponds to LC1, was S-nitrosylated in dissected somite strips and that S-nitrosylation of this band was prevented by an NOS inhibitor. Finally, they showed that PDI is expressed on mature astrocytes and that the growth cone collapsing activity of the grey matter of mature mammalian brain is blocked by inhibitors of PDI activity, suggesting that a similar mechanism operates in the CNS.

It is unfortunate that the chicken lacks suitable transgenetic tools, so well developed in mice, to demonstrate conclusively the role of PDI in segmental axon patterning. Nonetheless, the authors have done what is acceptable to show beyond reasonable doubt the role of PDI in segmental axon patterning and its mechanism of action in the developing chicken embryo. This is an important, far-reaching study. Many of the experiments are truly heroic given the very small quantity of tissue available. The results are well-controlled and logically presented and the paper is very well written. I have no major criticisms.

Reviewer #2:

This is a careful study in which a plethora of methodologies are used to dissect the molecular mechanisms by which cell surface protein disulphide isomerase underpins sensory axon segmentation of spinal neurons. It has important implications not just for developmental neuroscience but also for potential strategies to improve recalcitrant neuron growth in the mammalian central nervous system.

Overall the manuscript is well written and clearly organised into a sequence of logical experiments. It is evident that the authors have substantial expertise in the subject area. However, there are a few things that I would like to see addressed.

1) In terms of figure presentation, I think it would help enormously to have cartoons of the somite arrangements and the axon tracks coupled with the key molecular cues to set the stage for the study. That is, to describe the anatomy in a classical developmental biology/anatomy context. I don't think this is a minor point. *eLife* is a journal with a wide audience and the paper is written from the viewpoint of a hard-core developmental neuroscientist, which shows deep understanding of the specialist area, but which might make the paper inaccessible to a larger audience. A summary figure to set out the hypothesis in a first figure, coupled with a summary diagram in a final figure pulling together the various strands of the results would be incredibly useful. Both because of the wide range of specialist techniques used in the study and because there is a wider context in which the study could be interpreted- but there is a lot to digest from the text and the point is difficult to grasp in places.

2) The authors say that ….'co-injection of siRNA…rescued the normal segmented phenotype (Figure 2….)'. But looking at the data in the figure I think it is best to say that the phenotype was rescued partially.

3) A major point of the paper is that the authors have identified a growth cone collapse mechanism and they use a collapse assay (collected data are based on morphology) but there are no photographs of the growth cones in any of the figures. It would be useful to have images so the reader can get a feel for what 'collapse' looks like, especially with respect to soluble and contact mediated factors. For example, when the growth cones are challenged with soluble factors or liposomes: how do liposomes interact physically with filopodia in these cultures as compared to soluble factor influences on growth cone morphology (important because, as the authors state, a single filopodial contact can make all the difference)? They report time course data, so they must have time lapse images of growth cones as they collapse from which such measurements were made. These would be useful for the reader.

4) In a similar vein, given the importance of growth cone collapse and the role of the identified molecular mechanism on axon pathfinding I'm wondering why the authors did not include 'classical growth cone turning assay' or 'stripe assay' experiments. I am not necessarily requesting additional experiments, but if the authors have such data they would add to the study.

Reviewer #3:

In this interesting manuscript, Cook and colleagues provide a kind of closure to a long-standing question about patterning of metameric nerve growth in avian and mammalian embryos. Previous studies, many from the Keynes laboratory, showed that although both neural crest cells and motor and sensory axons traverse only the anterior half somite during outgrowth, the underlying molecular mechanisms that channel neural crest cell migration are distinct from those that channel axon extension. Here, they identify PDIA1/P4HB, a protein disulphide isomerase, as the cell surface activity required to prevent axons from entering the posterior half somite. They demonstrate that this PDI mediates axon repulsion by activating nitric oxide signaling, which elicits growth cone collapse. The also show that this PDI is the activity from adult brain extracts previously shown to mediate sensory axon growth cone collapse, and that this activity may emanate from astroglia.

This manuscript provides a satisfying answer to a question that has long been elusive. For the most part the data are very robust and well-documented. However, I suggest two types of improvements. First, other molecules, including ones described by the Keynes laboratory, are also implicated in preventing axons from entering the posterior half somite. This should be discussed. I think it would be very helpful to include a model figure that shows how PDI operates, as they describe in the Discussion, and also includes these other molecular mechanisms. Second, a number of figures could be improved.

1) Figure 1: It would be helpful to box the regions blown up in D-F.

2) Figure 1—figure supplement 1: It's very difficult to see what's going on in panel C. Asterisks or some other marker would help.

3) Figure 2: Is the loss of PDI random in these experiments? It looks like after siRNA it is gone in some somites and still present in others. Is this consistent? If so, how do you know whether it is effective?

4) Figure 2—figure supplement 2. It would be very helpful to show a blow up with comparisons for panel H.

[Editors' note: further revisions were suggested prior to acceptance, as described below.]

Thank you for submitting your article "Regulation of nerve growth and patterning by cell surface protein disulphide isomerase" for consideration by *eLife*. Your article has been reviewed by three peer reviewers, and the evaluation has been overseen by a Reviewing Editor and Marianne Bronner as the Senior Editor The following individual involved in review of your submission has agreed to reveal their identity: Alun M Davies (Reviewer #1).

The reviewers have discussed the reviews with one another and the Reviewing Editor has drafted this decision to help you prepare a revised submission.

Summary:

All three reviewers were enthusiastic about your study dissecting the molecular mechanisms by which the cell surface protein disulphide isomerase contributes to segmentation of sensory axons of spinal neurons. The study should appeal both to developmental biologists and those who study neuronal pathologies.

Reviewer 1 stated that "this is an excellent study that has been further improved by the revisions. The authors are to be congratulated on their achievements" and had no further comments at this stage. Reviewers 2 and 3 stated that the manuscript has been improved, and that you addressed the suggestions made and issues raised.

Essential revisions:

While reviewers 2 and 3 found that the changes to the figures, especially Figure 1, clarified the experimental outcomes and interpretation, there were three remaining amendments suggested:

1) In the figure legend for new Figure 1, you refer to '…mixed spinal nerves (yellow) but there is no yellow in the figure. Neither the arrow pointing to the nerve nor the nerve itself is yellow, so you may want to amend this.

2) New Figure 5 is also helpful, but reviewer 3 found the large red arrow at the bottom confusing. Is this meant to indicate retraction? If so, that should be described in the legend.

3) Reviewer 3 wonders why the you chose not to include an image of a collapsed growth cone, as suggested originally by reviewer 2. This would make the article more accessible to readers, without them having to look at additional references.

---

## [Author Response]

Essential revisions:All of the revisions requested below are essential, and none call for performing new experiments.Reviewer #1:The restriction of the growth of sensory and motor axons to the anterior half of each somite-derived sclerotome, generating the segmented patterning of the early PNS, and the demonstration that this is due to contact repulsion exerted by the posterior sclerotome cells is one of the first and classic examples of this fundamental axon guidance mechanism. […] The results are well-controlled and logically presented and the paper is very well written. I have no major criticisms.

The reviewer raises no major criticisms, and regards the study as a 'major and important contribution to our understanding' of segmental axon patterning. We note the comment on our use of the chick embryo rather than mouse, which stems from earlier embryo grafting experiments that defined this biological system. Relevant here, we have added a new comment in the Discussion (third paragraph) regarding the mouse mutant screen published by Samuel Pfaff and colleagues, and a reference to this work (Bai et al., 2011). Their *Columbus* mutant has an impressive spinal axon phenotype indicative of a loss of somite polarity (P-to-A). This likely arose because the mutant gene, *Presenilin-1*, regulates Notch signalling that is required upstream to generate P-half somite polarity.

Reviewer #2:Overall the manuscript is well written and clearly organised into a sequence of logical experiments. It is evident that the authors have substantial expertise in the subject area. However, there are a few things that I would like to see addressed.1) In terms of figure presentation, I think it would help enormously to have cartoons of the somite arrangements and the axon tracks coupled with the key molecular cues to set the stage for the study. That is, to describe the anatomy in a classical developmental biology/anatomy context. I don't think this is a minor point. eLife is a journal with a wide audience and the paper is written from the viewpoint of a hard-core developmental neuroscientist, which shows deep understanding of the specialist area, but which might make the paper inaccessible to a larger audience. A summary figure to set out the hypothesis in a first figure, coupled with a summary diagram in a final figure pulling together the various strands of the results would be incredibly useful. Both because of the wide range of specialist techniques used in the study and because there is a wider context in which the study could be interpreted- but there is a lot to digest from the text and the point is difficult to grasp in places.

Regarding the need for a diagram of the developmental anatomy, we entirely agree with the reviewer that this will help readers unfamiliar with this system, and have added the requested summary figure (new Figure 1A) to clarify the anatomical relationships.

2) The authors say that ….'co-injection of siRNA…rescued the normal segmented phenotype (Figure 2….)'. But looking at the data in the figure I think it is best to say that the phenotype was rescued partially.

We take the point that we did not see complete rescue and have stated that the phenotype was rescued 'partially' as suggested (subsection “csPDI mediates spinal nerve patterning in vivo”).

3) A major point of the paper is that the authors have identified a growth cone collapse mechanism and they use a collapse assay (collected data are based on morphology) but there are no photographs of the growth cones in any of the figures. It would be useful to have images so the reader can get a feel for what 'collapse' looks like, especially with respect to soluble and contact mediated factors. For example, when the growth cones are challenged with soluble factors or liposomes: how do liposomes interact physically with filopodia in these cultures as compared to soluble factor influences on growth cone morphology (important because, as the authors state, a single filopodial contact can make all the difference)? They report time course data, so they must have time lapse images of growth cones as they collapse from which such measurements were made. These would be useful for the reader.

Regarding the presentation of images of collapsed growth cones, we have clarified this in the Materials and methods section (subsection “Growth cone collapse assays”). We have made it clear that collapse is defined morphologically as loss/conversion of the entire growth cone structure (i.e. both filopodia and lamellipodia) to a needle-like point, with no more than two filopodia. We have additionally referred to images of collapsed growth cones in a previous publication from our lab (Manns et al., 2012, Figure 1K, L). These show that using phase-contrast optics to visualise growth cones, as we have also done here, compares favourably with phalloidin staining of growth cones. Regarding the reviewer's further comment on liposomes, as a control we use liposomes that have not incorporated any proteins, and routinely we find they do not cause collapse beyond the background levels also seen after addition of phosphate-buffered saline.

4) In a similar vein, given the importance of growth cone collapse and the role of the identified molecular mechanism on axon pathfinding I'm wondering why the authors did not include 'classical growth cone turning assay' or 'stripe assay' experiments. I am not necessarily requesting additional experiments, but if the authors have such data they would add to the study.

Regarding use of the stripe- and turning assays, we did not investigate these for purification purposes; they would have been far more time-consuming and thereby less suitable for the large number of multiple assays required for biochemical purification and characterization.

Reviewer #3:This manuscript provides a satisfying answer to a question that has long been elusive. For the most part the data are very robust and well-documented. However, I suggest two types of improvements. First, other molecules, including ones described by the Keynes laboratory, are also implicated in preventing axons from entering the posterior half somite. This should be discussed. I think it would be very helpful to include a model figure that shows how PDI operates, as they describe in the Discussion, and also includes these other molecular mechanisms. Second, a number of figures could be improved.

– Other candidate repellent molecules have been discussed as requested (Discussion, together with associated new references). We have also added a sentence regarding previous studies indicating a role for NO signalling in synapse elimination during brain development.

– Also as requested we have added a new figure (Figure 5) to illustrate the proposed operation of csPDI. We do agree that addition of the other candidate molecules is very helpful, and the relevant information is now included in the Discussion as noted above. We would prefer not to add these other candidate molecules to this figure, feeling it would become cluttered with too much information as a result.

1) Figure 1: It would be helpful to box the regions blown up in D-F.

Figure 1D-F: boxes have been added as requested.

2) Figure 1—figure supplement 1: It's very difficult to see what's going on in panel C. Asterisks or some other marker would help.

Figure 1—figure supplement 1C: arrows have been added for greater clarity.

3) Figure 2: Is the loss of PDI random in these experiments? It looks like after siRNA it is gone in some somites and still present in others. Is this consistent? If so, how do you know whether it is effective?

Figure 2 – regarding whether loss of PDI is random. Panel A shows that fluorescein-labelled siRNA was delivered into several consecutive somites after a typical injection experiment. Therefore, in respect of siRNA distribution in a contiguous row of somites, we did not predict randomly-distributed loss of PDI from segment-to-segment. Abnormal outgrowth patterns were seen in consecutive segments, as shown in panels C and D (on the right in each image); these panels also show normal segmented outgrowth in more posterior segments (to the left in each image), presumably where siRNA delivery was diminished compared with the abnormal segments. The legend has been modified accordingly to make this point clearer.

In the case of a single somite we did not expect the knockdown to affect every cell within it, nor did we see evidence for this. Figure 2—figure supplement 2J and K show a scattered distribution of sclerotome cells expressing PDI (J) and the FLAG M1 epitope (K). In view of the cell-surface-localized nature of the contact-based repulsion mechanism, we would expect such somites to elicit the sprouting defects as described. We find no evidence that P-half-somite cells secrete an axonal repellent (Keynes et al., 1997), so cells that escape the siRNA would not be expected to compensate knocked-down cells in the same somite. We prefer not to lengthen the Discussion with these arguments but will do this if so required.

4) Figure 2—figure supplement 2. It would be very helpful to show a blow up with comparisons for panel H.

Figure 2—figure supplement 2H: the magnification has been increased.

[Editors' note: further revisions were suggested prior to acceptance, as described below.]

Essential revisions:While reviewers 2 and 3 found that the changes to the figures, especially Figure 1, clarified the experimental outcomes and interpretation, there were three remaining amendments suggested:1) In the figure legend for new Figure 1, you refer to '…mixed spinal nerves (yellow)’ but there is no yellow in the figure. Neither the arrow pointing to the nerve nor the nerve itself is yellow, so you may want to amend this.

We have deleted '(yellow)' from the figure legend as we feel this is redundant since the 'mixed' spinal nerve is described in the legend and identified with an arrow in the figure.

2) New Figure 5 is also helpful, but reviewer 3 found the large red arrow at the bottom confusing. Is this meant to indicate retraction? If so, that should be described in the legend.

Yes, the large red arrow is meant to indicate retraction and this is now clarified in the legend.

3) Reviewer 3 wonders why the you chose not to include an image of a collapsed growth cone, as suggested originally by reviewer 2. This would make the article more accessible to readers, without them having to look at additional references.

We have revised Figure 3—figure supplement 3 by incorporating an additional panel (new panel B) showing spread and collapsed growth cones in the PDI+GSNO experiment. We have also made the necessary accompanying minor changes to the main text and the figure legend to accommodate this amendment.